# Hallucination of closed repeat proteins containing central pockets

Linna An [1,2,4] ✉, Derrick R. Hicks[1,2,4], Dmitri Zorine [1,2,4], Justas Dauparas[1,2], Basile I. M. Wicky [1,2], Lukas F. Milles [1,2], Alexis Courbet [1,2,3], Asim K. Bera [1,2], Hannah Nguyen [1,2], Alex Kang[1,2], Lauren Carter[1,2] & David Baker [1,2,3] ✉

In pseudocyclic proteins, such as TIM barrels, β barrels, and some helical transmembrane channels, a single subunit is repeated in a cyclic pattern, giving rise to a central cavity that can serve as a pocket for ligand binding or enzymatic activity. Inspired by these proteins, we devised a deep-learning-based approach to broadly exploring the space of closed repeat proteins starting from only a specification of the repeat number and length. Biophysical data for 38 structurally diverse pseudocyclic designs produced in *Escherichia coli* are consistent with the design models, and the three crystal structures we were able to obtain are very close to the designed structures. Docking studies suggest the diversity of folds and central pockets provide effective starting points for designing small-molecule binders and enzymes.

Native cyclic repeat proteins have a broad array of biological functions. For example, the triosephosphate isomerase (TIM) barrel[1], which consists of eight α/β repeats that close to form an eight-stranded β-barrel surrounded by an outer ring of helices, is the most prevalent protein fold for enzymes. Single-chain cyclic structures formed by repeating units have considerable advantages: at the center is a pocket into which side chains from each repeat unit extend, and because these structures are single chains, the sequence lining (and local structure) can be fully asymmetric. De novo protein design has been used to create repeat proteins that do not close[2,3], and closed TIM barrels[4], parametric bundles[5], and all α-helical toroids[6]. However, a general method for broadly sampling repeating cyclic structures without specifying the overall architecture or lengths and positions of the secondary structures has thus far been missing.

We sought to develop a general approach to overcome these limitations, to enable the generation of a wide range of new cyclic-protein scaffolds with central cavities from which small-molecule binders and enzymes can be designed. We reasoned that recently developed deep-network-based protein hallucination methods[7], which optimize sequences for folding to specific structures without requiring specification of what the structure is, could be extended to broadly sample

cyclic repeat protein structure space given only the repeat-unit length and the number of repeats.

## Results

### Hallucination and sequence design of pseudocycles

We developed a sequence space Markov Chain Monte Carlo (MCMC) optimization protocol (Fig. 1a), which, given the length (*L*) and number of repeating units (*N*), first generates a random amino acid sequence of length *L* and tandemly repeats it *N* times. We sampled *N* from 2 to 7, and *L* from 15 to 78, with a maximum protein length of 156 amino acids. The protocol then optimizes this sequence by making one to three random amino acid substitutions at a random position in one repeat unit, propagating these substitutions to all repeat units, evaluating the extent to which the sequence encodes a cyclic-repeating-protein structure, and finally accepting or rejecting the substitutions according to the standard Metropolis criterion. To evaluate sequence folding to a cyclic structure, we used AlphaFold2 (ref. 8) (AF2), with a single sequence as an input and three recycling stages to predict the structure, and we subsequently evaluated the extent of closure by extrapolating helical parameters from the rigid-body transformations between successive repeat units: closed structures are those with near-zero rise

[1]Department of Biochemistry, University of Washington, Seattle, WA, USA. [2]Institute for Protein Design, University of Washington, Seattle, WA, USA. [3]Howard Hughes Medical Institute, University of Washington, Seattle, WA, USA. [4]These authors contributed equally: Linna An, Derrick R. Hicks, Dmitri Zorine. ✉e-mail: linnaan@uw.edu; dabaker@uw.edu

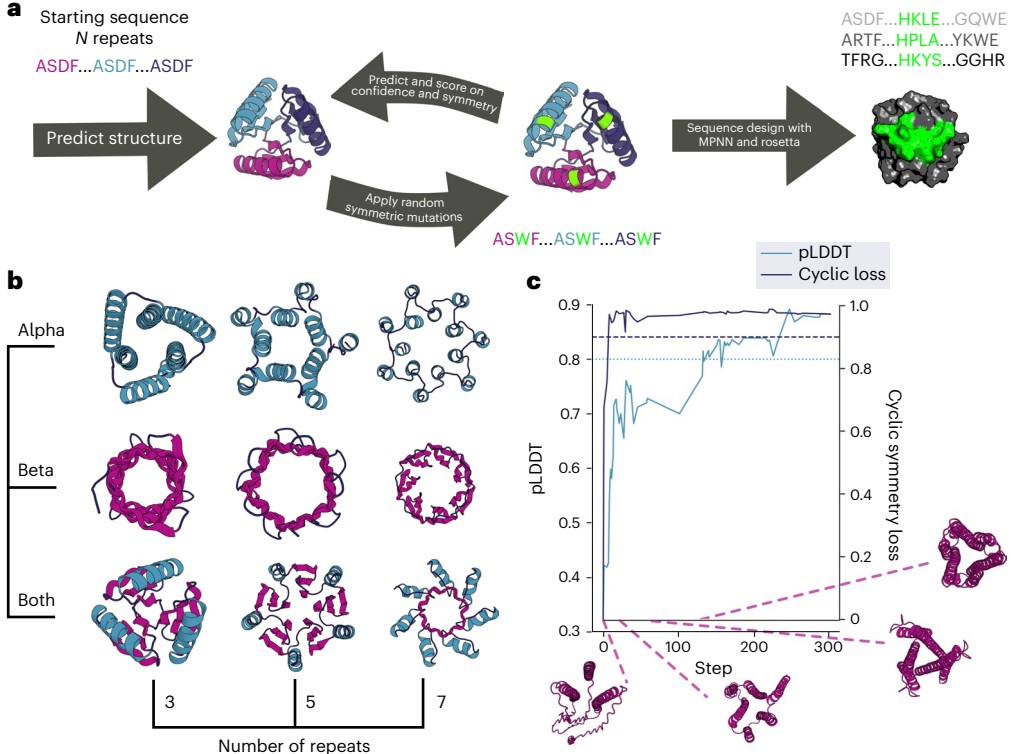

**Fig. 1 | Pseudocyclic protein design. a**, Schematic representation of the scaffold hallucination and design pipeline. **b**, Selected output proteins featuring 3, 5, or 7 repeats, and all-α (Alpha), all-β (Beta), or mixed α/β (Both) topologies. **c**, Representative design trajectory showing the optimization of predicted local distance difference test (pLDDT, teal) and cyclic loss (dark blue) over 300 steps, with dashed lines indicating our selected score cutoffs. Protein structure cartoons are snapshots at indicated steps in the trajectory; loop, sheet, and helix regions are colored in dark blue, magenta, and teal, respectively.

along the helical axis, and rotation of 360 / N degrees about the helical axis. To guide the MCMC trajectories, we supplemented the closure score with AF2 confidence prediction metrics (see 'Protein generation and sequence design pipeline' in Methods). We found that, after only a few hundred steps (see Extended Data Fig. 1), most MCMC trajectories optimizing for this combined score converge on sequences that are predicted to fold with high confidence into closed cyclic structures (Fig. 1b). Because of steric exclusion of the closed cyclic structures, individual structural elements avoided clashing with the symmetry axis and formed cavities of various sizes in the center.

Although this cyclic hallucination procedure generated a wide array of new cyclic backbones, the actual amino acid sequences contained sub-optimal features, such as large hydrophobic surface patches and poor secondary structure sequence agreement (Extended Data Fig. 2). A limitation of the hallucination procedure, as with any activation maximization procedure that optimizes over the inputs to a neural network, is generation of adversarial examples by overfitting. Hallucination studies on cyclic oligomer design have shown that, although AF2-generated sequences were rarely soluble, redesign of the hallucinated backbones with ProteinMPNN yielded soluble proteins with the desired structures[9]. Because the backbones are intended for ligand-binder design, and most ligands are not symmetric, we used ProteinMPNN[10], which we gave the hallucinated backbones (see 'Protein generation and sequence design pipeline' in Methods), to design new sequences without requiring sequence-repeat symmetry, which resulted in sequence-asymmetric final designs (Fig. 1c). Finally, we used RoseTTAfold[11] (RF) and AF2 to evaluate the extent to which the designed sequences encoded the intended structures (Extended Data Fig. 3).

We obtained a total of 21,021 designs that were strongly predicted to fold to the intended structures. We refer to these designs as 'pseudocycles' because their backbones have near cyclic symmetry (except for the break between the carboxy and amino termini) but the sequence is asymmetric. The 21,021 designed pseudocycles span a very wide range of topologies containing all α, α/β, and all β subdomains (see Figs. 1b, 2, and 3 and Extended Data Fig. 4). In some of the designs, the repeat units form compact domains that interact with neighboring units through relatively small interfaces; in others, the repeat units are more intertwined (Fig. 3). To evaluate how thoroughly our calculations sample the space of possible pseudocycles, we first reduced the structural redundancy (see 'Protein clustering' in Methods), then we randomly selected 10 subsets of designs with 500, 700, 1,000, or 5,000 members from the redundancy-reduced pseudocycle sets. For each subset, we determined the fraction of designs with structures that were very different from any other member of the subset (template modeling (TM) score < 0.45, Extended Data Fig. 5). For the smallest subsample sizes, the fraction of singleton scaffolds approached 20%; with increasing subsample sizes, this fell to below 2.5% (Extended Data Fig. 5a). Thus our structure-generation-by-hallucination procedure identified almost all pseudocycle solutions that pass our selection criteria multiple times, suggesting that our set of 21,021 designs fairly comprehensively covers the space of pseudocycles that can be generated using our approach.

## Experimental characterization of selected pseudocycles

We selected 96 designs that varied in the number of repeat units, length, and secondary structure composition for experimental characterization, focusing on designs containing designable pockets and folds that are rare in the Protein Data Bank (PDB) (Fig. 2). These proteins have sequences and structures that are different from those in the PDB, with a median Basic Local Alignment Search Tool (BLAST) expect (E) value of 0.018 and TM scores between 0.33 and 0.87 (average value,

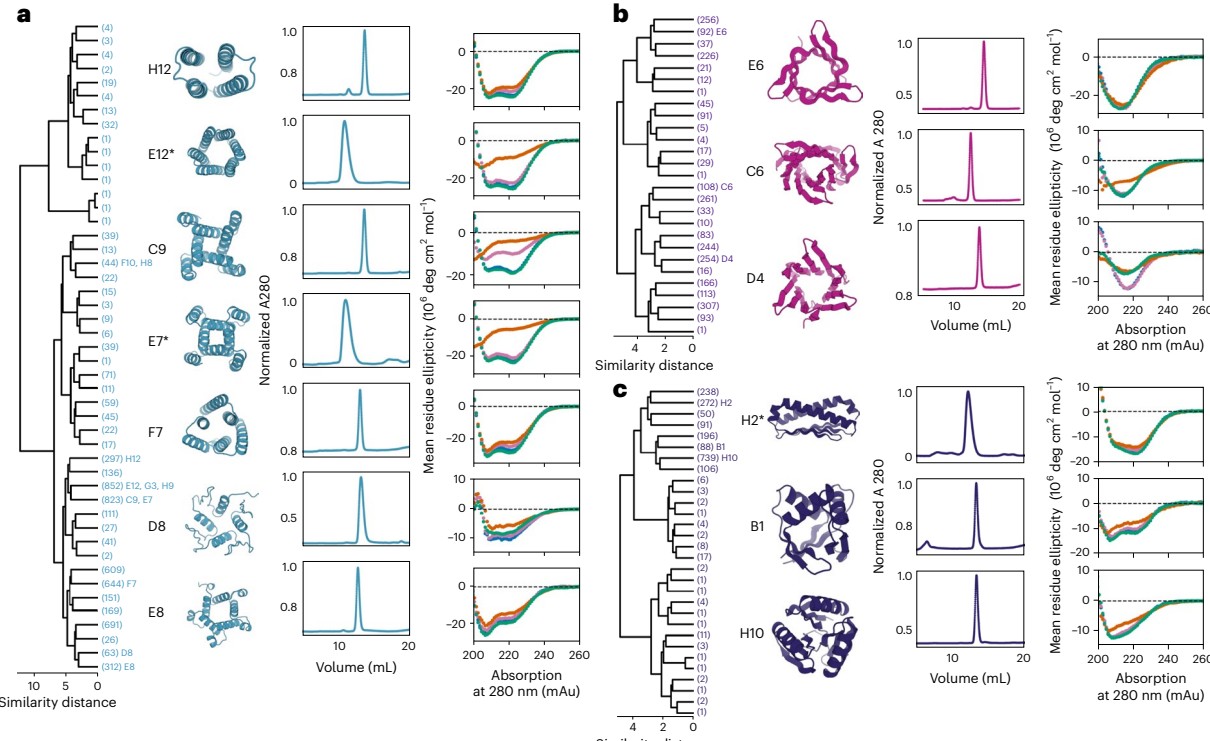

**Fig. 2 | Biophysical characterization. a–c**, First panel, hierarchical clustering of designed pseudocycles, the x axes represent relative structural similarity distance, the smaller the distance, the more similar the two structures are. The number of sub-branches are indicated in brackets. Second panel, diagrams of designs selected for experimental characterization; identifiers indicate position in dendrograms. Third panel, size-exclusion chromatography (SEC) trace with normalized absorption at 280 nm (A280) plotted on the y axis against the elution volume plotted on the x axis. Protocols are described in the Methods ('Expression and purification of selected proteins'). Proteins prepared following protocol 1 are marked with an asterisk. Fourth panel, CD spectra at different temperatures (25 °C in blue, 55 °C in orange, 95 °C in pink, followed by refolding at 25 °C in green). **a**, α-helical topologies (colored teal); **b**, β-sheet topologies (colored magenta); **c**, mixed α/β topologies (colored dark blue).

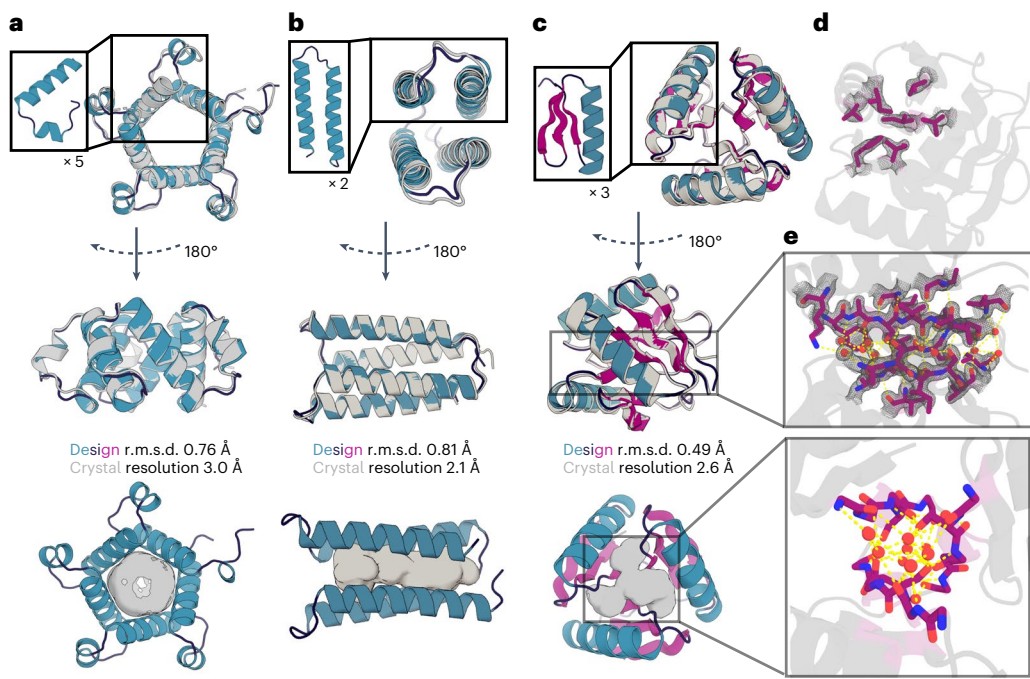

**Fig. 3 | X-ray crystal structures of the designs are very close to the computational models. a–e**, Crystal structures of the five-repeat design E8 (**a**), two-repeat design H12 (**b**), and the three-repeat design H10 (**c–e**) are shown as gray cartoons; the loop, sheets, and helix of the design are shown in dark blue, magenta, and teal, respectively. Central pockets in the designs are shown as gray spheres (**a–c**). The secondary structure interface (**d**) and the center-water-mediated hydrogen bond network (**e**) of the refined crystal structure of design H10 are shown using sticks. The electron density map of the interface and the center hydrogen bond network and water are shown as gray mesh. In **d** and **e**, the oxygen, nitrogen, and carbon are colored in red, blue, and magenta, respectively; hydrogen bond networks are shown as yellow dashed lines, and water molecules as red spheres.

**Table 1 | Crystallographic data collection and refinement statistics**

| | H10 (8FJF) | H12 (8FJG) | E8 (8FJE) |
|---|---|---|---|
| **Data collection** | | | |
| Space group | $P2_12_12_1$ | $P2_12_12_1$ | $P22_12_1$ |
| Cell parameters | | | |
| a,b,c (Å) | 34.00, 44.28, 78.12 | 30.45, 46.59, 73.45 | 47.85, 65.76, 87.43 |
| α, β, γ (°) | 90, 90, 90 | 90, 90, 90 | 90, 90, 90 |
| Resolution (Å)[a] | 38.53–1.60 (1.65–1.60) | 24.08–2.13 (2.21–2.13) | 52.56–3.0 (3.3–3.0) |
| Unique reflections | 16,171 (1,573) | 6,175 (546) | 5,818 (1,428) |
| $R_{merge}$ | 0.1616 (1.630) | 0.1677 (1.026) | 0.2032 (1.24) |
| $R_{pim}$ | 0.0470 (0.4726) | 0.0512 (0.3653) | 0.06071 (0.3464) |
| $I/\sigma(I)$ | 10.22 (0.97) | 11.03 (0.92) | 7.95 (2.48) |
| Wilson $B_{factors}$ (Å$^2$) | 31.28 | 58.99 | 92.27 |
| $CC_{1/2}$ | 0.996 (0.579) | 0.992 (0.505) | 0.993 (0.903) |
| Completeness (%) | 99.88 (99.87) | 98.26 (91.00) | 98.76 (99.72) |
| Redundancy | 12.9 (12.9) | 12.2 (8.0) | 12.7 (13.6) |
| **Refinement** | | | |
| Resolution (Å) | 38.53–1.60 (1.65–1.60) | 24.08–2.13 (2.21–2.13) | 52.56–3.0 (3.3–3.0) |
| No. reflections | 16,156 (1,573) | 6,090 (546) | 5,818 (1,428) |
| $R_{work}$ / $R_{free}$ | 0.1961 (0.3347) / 0.2272 (0.3643) | 0.2577 (0.4248) / 0.2851 (0.4319) | 0.2660 (0.3250) / 0.2962 (0.3625) |
| No. atoms | | | |
| Protein | 978 | 878 | 2,312 |
| Solvent | 62 | 5 | 0 |
| Ramachandran favored/allowed /outlier (%) | 100.00/0.00/0.00 | 98.04/1.96/0.00 | 95.52/4.48/0.00 |
| R.m.s. deviations | | | |
| Bond lengths (Å) | 0.011 | 0.008 | 0.002 |
| Bond angles (°) | 1.05 | 0.91 | 0.45 |
| $B_{factors}$ (Å$^2$) | | | |
| Protein | 39.76 | 69.01 | 94.70 |
| Solvent | 43.29 | 62.39 | n/a |

[a]Statistics for the highest-resolution shell are shown in parentheses.

0.54; Supplementary Table 1). We found that, following expression in *E. coli*, 81 of the 96 designs were soluble, and 38 of these 81 soluble designs were well-expressed and had circular dichroism (CD) spectra, indicating that the proteins were well-folded with overall secondary structure content consistent with the design models (Fig. 2 and see Extended Data Figs. 6 and 7). Seventeen of these 38 designs were monomeric and monodisperse; another 15 were polydisperse with a majority monomeric population (Fig. 2 and see Extended Data Figs. 8 and 9)[12].

We were able to solve crystal structures of three designed pseudocycles. In all three cases, the crystal structures show closed repeat structures very similar to the computational design models (Fig. 3 and Table 1). The first structure is a helical bundle with pseudo-$C_5$ symmetry formed by replication of a helix-turn-helix repeat (Fig. 3a and Extended Data Fig. 10). The design model was very accurate, with a Cα r.m.s. deviation (r.m.s.d.) of 0.6 Å with respect to the solved structure, and was very different from any structure in the PDB (the closest structure, 3WFB, chain B, has a TM score of 0.392). The outside short helix stubs form hydrophobic interfaces with the two neighboring long helices, and the hydrophobic side chains lock the helices while leaving a big pocket in the middle (see Extended Data Fig. 10). The second structure is a simple four-helix bundle with pseudo-$C_2$ symmetry formed by duplication of a helix hairpin repeat (Fig. 3b); the closest structure in

the PDB (5OXF) has a TM score of 0.65. The design model is also very accurate, with a Cα r.m.s.d. of 0.8 Å with respect to the solved structure. The third structure is a more complex pseudo-$C_3$-symmetric protein with a repeated EEHE fold (Fig. 3c–e). The interface between repeat units contains seven buried hydrophobic residues contributed by a helix-strand motif from one repeat packing into a groove formed by three strands from the next repeat unit (Fig. 3d). A water-mediated hydrogen bonding network is formed between the three center strands and water molecules occupying the central cavity (Fig. 3e). The design model was again very accurate, with a Cα r.m.s.d. of 0.5 Å with respect to the solved structure, and was very different from any structure in the PDB (closest TM score of 0.38 to 1U7Z). All three solved structures, like the vast majority of our designs, contain central pockets that can be used to design small-molecule binders and enzymes (Fig. 3a–c).

**Small-molecule docking for pseudocycles and other scaffolds**

To investigate the potential of the designs for scaffolding ligand-binding pockets, for each of 9,838 cluster centers (see Supplementary Fig. 1 and 'Protein clustering' in Methods), we carried out rotamer interaction field (RIF) docking[13] and pocket-design calculations with 19 ligands with diverse sizes, shapes, and chemical properties (Supplementary Fig. 2 and 'Ligand docking to pseudocycles, NTF2, and native proteins'

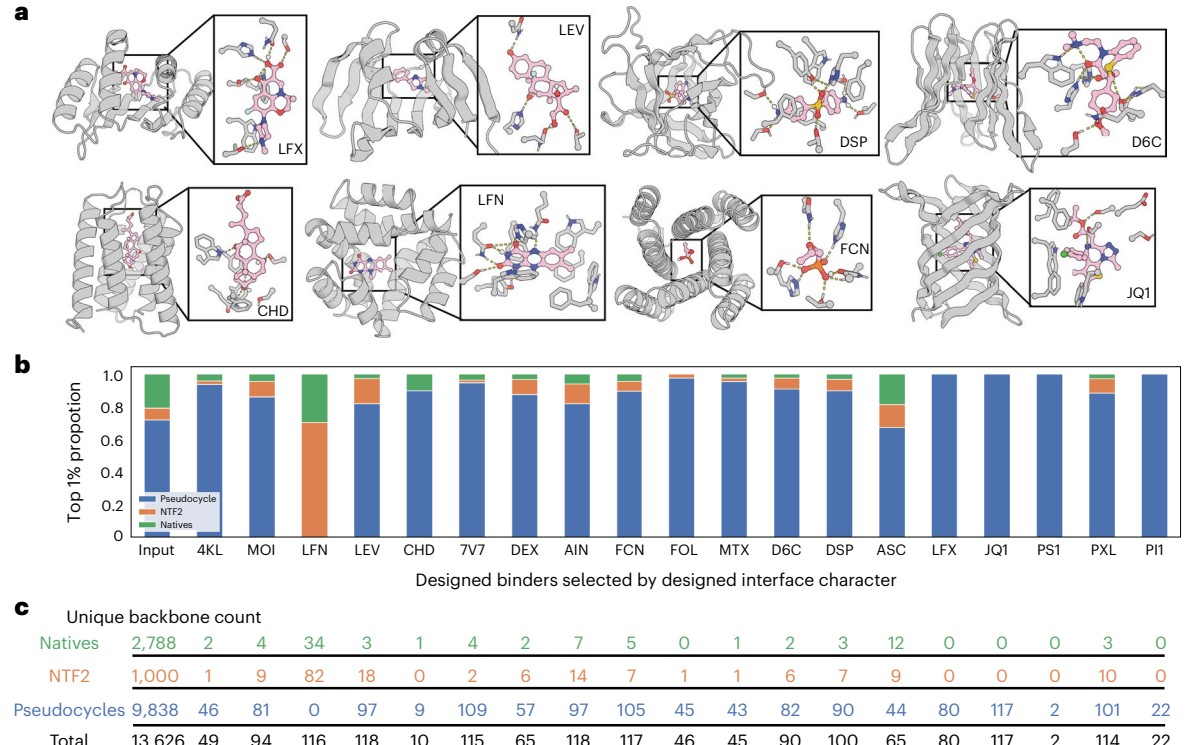

**Fig. 4 | The central pockets can accommodate a wide range of small-molecule ligands. a**, Examples of computationally designed binding interactions for diverse ligands in pockets of diverse pseudocyclic scaffolds. Designed proteins are shown as gray cartoons and sticks, and the ligands are shown as pink sticks. Oxygen, nitrogen, phosphorus, and chlorine elements are colored in red, blue, orange, and green, respectively. **b**, Barplots showing the composition of input scaffolds (pseudocyclic designs, native structures from the PDBBind database, and designed NTF2s from Basanta et al.[16]) and subsequent composition of the best-ranked small-molecule binders (each scaffold can contribute up to 30 binders) for diverse ligands. **c**, The numbers of unique backbone scaffolds selected on the basis of the top 1% designed interface character from each type of scaffold are listed. LFX, levofloxacin; LEV, lenvatinib; DSP, dapsone; D6C, diltiazem; FCN, fosfomycin; LFN, lumiflavin; CHD, cholic acid.

in Methods)[14]. For each of the 19 ligands, we also carried out RIF docking and design calculations for 2,787 single-chain native small-molecule binding proteins from the PDB (PDBBind[15]) and 1,000 previously published de-novo-designed NTF2-like proteins[16] (see 'Ligand docking to pseudocycles, NTF2, and native proteins' in Methods). For each ligand docking onto a pseudocycle or native protein scaffold, the scaffold sequence at the small-molecule interface was optimized for high-affinity binding using the Rosetta sequence design suite[17], and the scaffolds that were most suitable for each ligand were picked on the basis of predicted ligand-binding energy, shape complementarity, and related docking-quality metrics[14]. Examples of designed binding sites for several diverse ligands to their most-suitable pseudocyclic proteins are shown in Figure 4a. We found that, for most ligands, the binding sites that could be most easily designed were obtained using the pseudocycle scaffolds (Fig. 4b,c and 'Ligand docking to pseudocycles, NTF2, and native proteins' in Methods), likely because of the great variety of binding-site shapes (see Supplementary Fig. 1) and sizes, and the many Cα-Cβ vectors pointing into the pocket, which together enable design of plausible binding sites for almost any ligand.

## Discussion

Our results further illustrate the power of deep network hallucination to explore the space of possible protein structures given only general specifications of structural features—in this case, the number and length of the repeating units, and the constraint that the repeat units close on themselves to form an overall structure with cyclic symmetry. The approach generates a wide variety of structures strongly encoded by their amino acid sequences (as evaluated with RF and AF2,

and further indicated by the close agreement between the crystal structures and design models), with between 2 and 7 repeat units (*N*), repeat lengths of 15 to 78 (*L*), and all α-, α/β-, and all β-folds. The sequences of the designs are unrelated to those of naturally occurring proteins, and although some structures resemble naturally occurring proteins, many have novel tertiary structures. Compared with previous Rosetta-based scaffold-generation pipelines[3,16], our pipeline can freely sample structure space and generate widely diverse protein architectures. Our pipeline also has substantially higher efficiency at both the computational and experimental levels. Out of roughly 28,000 generated pseudocycles, 21,021 (73.84%) had Rosetta, AF2, and RF metrics predictive of folding. By comparison, for the simpler problem of helical repeat protein design, the success rate for a previous Rosetta-based protocol was only 11,243/2,880,000 = 0.39% (ref. 3). At the experimental level, 84.4% (81/96) of the designs from our pipeline are highly soluble, compared with 17/64 = 26.6% for previous Rosetta-based design of small-molecule-binding (NTF2) scaffolds[16] (the solubility rate for Rosetta-based design of helical repeat proteins lacking cavities is 74/83 = 89.2%)[3].

The de novo design of small-molecule ligand-binding proteins and enzymes is still in its infancy. Two approaches have previously been used. The first is redesigning naturally occurring proteins[18], which is often limited by the relatively low stability of native proteins and the complexity of both the structure and sequence–structure relationships: such redesigns are often unstable or have unpredictable structural changes. The second approach is to start from robust de novo-designed scaffolds that lack features that are difficult to control, such as long loops, and that have better-understood

sequence–structure relationships. This approach has been limited by the lack of diversity in available de novo-designed scaffolds[13,16,19] when compared with the diversity of structures in the PDB. The large set of pseudocycles with central binding pockets described here combines the diversity of native protein scaffolds with the stability and robustness of de novo-designed proteins, and our RIF docking and Rosetta design calculations suggest that the designed pseudocycles provide better starting scaffolds for small-molecule binder design than do either native structures or previously designed de novo NTF2s. Future work will focus on designing and experimentally characterizing small-molecule binding proteins and enzymes using these scaffolds.

## Online content

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

## Methods

### Protein generation and sequence design pipeline

Initial models were derived through AF2 prediction of randomly generated amino acid sequences. Sequence space was traversed through substitutions (one to three substitutions at a time) propagated in repeat units followed by evaluation of the predicted structure of the modified sequence. Cyclic character was evaluated as helical rise near zero and per-unit rotation near 360 / $N$ degrees. For each predicted model, the difference between the computed values and the ideal values was calculated and then rescaled logistically to a score between 0 and 1.

The closure score is a score from 0 to 1 (0 indicates perfect closure; 1 indicates no rotation within the unit) that is a linear combination of rescaled delta rise (the value change of rise) and rescaled delta rotation (the value change of rotation). Delta rise and delta rotation were computed by extrapolating a screw axis from smoothed relative transforms between repeat units. Relative transforms between matching repeats were derived from protein backbone positions, using the nitrogen, α-carbon, and carboxyl carbon to define the local coordinate plane for each residue position, and the homogenous relative transform between two coordinate frames was then derived. Relative transforms were averaged by averaging the quaternions corresponding to the relative transforms and directly computing the mean translation vector from the relative transforms. This smoothed transform was then used as a proxy for a rigid-body transform that represents the relationship of repeat units. A helical axis was derived from that transform, as were rise along and rotation about that axis.

The ideal rise for a cyclic repeat is 0 and an ideal rotation is 360 / $N$ degrees. The predicted structure's delta rise and delta rotation, relative to the ideal values, were rescaled logistically to yield values between 0 and 1, with midpoints at desirable delta values.

The logistic rescaling function used was:

$$\frac{1}{1 + e^{-s(x-m))}}$$

Where $s$ is the slope factor, $x$ is the delta value, and $m$ is the logistic scale midpoint. $m$ and $s$ for rotation were 4 and 1.5, respectively; those values for rise were 2 and 2, respectively. The mean between these rescaled values was used as a score for closure quality.

After generating initial pseudocycle scaffolds, we used ProteinMPNN[10] to design the sequences as asymmetric monomeric proteins and used AF2 and RF to generate structural models for the new sequences.

Visual inspection of the ProteinMPNN-designed pseudocycle models suggested that many designed proteins had large areas of surface-apolar residues (Extended Data Fig. 2). To resolve this issue, we used the Rosetta sequence design suite[17] to perform a redesign of the pseudocycles of their surface residues. For each pseudocycle model, we generated 100 sequences using ProteinMPNN[10], and we used these newly generated sequences to generate a scaffold-specific Position Specific Score Matrix (PSSM) file. Using Rosetta sequence design (FastDesign), surface residues of the pseudocycle models with a high spatial aggregation propensity[20] (SAP) score were selected, designed with the non-hydrophobic amino acid preference provided by the PSSM file, and scored with Rosetta metrics. These sequences were then used to predict new pseudocycle structures with AF2 and RF. We chose to use the AF2 rank1 model (which had the highest plDDT of the five available AF2 models). We collected AF2 metrics, including plDDT and predicted Template Modeling score (pTM); RF metrics, including plDDT, categorical cross-entropy (CCE), and Kullback–Leibler divergence (KL divergence); and the r.m.s.d. of the predicted structure with respect to the original design model (Extended Data Fig. 3).

To generate the final pseudocycle list for further design applications, we examined Rosetta, AF2, and RF metrics, and visually inspected the models. We removed the models that were predicted to fold into scaffolds with a Cα r.m.s.d. over 2 Å by AF2 with respect to the original model, and generated a finalized pseudocycle list consisting of 21,021 designed proteins.

The models of all 9,838 unique scaffolds and the sequences and the design models of the 96 characterized designs have been uploaded to GitHub (https://github.com/LAnAlchemist/Psedocycles_NSMB.git).

The scaffold-generation scripts have also been uploaded to GitHub (https://github.com/dmitropher/af2_multistate_hallucination.git).

### TMalign method

We used PyRosetta[21] to calculate the average TMscore between two input proteins by averaging the TMscore obtained when each of the PDB models was treated as the reference PDB model for sequence-length normalization.

### TMalign to natives

We curated a set of high-resolution (<1.8 Å resolution) structures without sequence redundancy (using MMseq[22]) from the PDB, which yielded 6,111 structures. We then ran TMalign as previously described to find designs with similar native structures (see 'TMalign method' in Methods).

### mTM-align of 96 characterized designs against the PDB

We used the mTM-align server[23] to compare the designs we characterized with the whole PDB, selecting the top database hit for TMalign comparison.

### Protein clustering

All scaffolds were first grouped on the basis of their initial symmetry number (2–7), and were then grouped on the basis of their repeat-unit secondary structural assignment based on Define Secondary Structure of Proteins algorithm; we then applied AgglomerativeCluster (from scikit-learn[24]) on an all-by-all matrix of TMalign scores within each group. Cluster count was selected by checking the cluster statistics of various cluster sizes: the final cluster number was chosen by minimizing the proportion of clusters with less than two members (singletons) with respect to the joint constraints of maintaining a high mean TMscore within each cluster and a low s.d. of intra-cluster TMscores. The final clustering yielded a mean intra-group TMscore of 0.88 (not including singleton clusters), with only 9.42% singleton clusters.

### Ligand docking to pseudocycles, NTF2, and native proteins

The pocket residues of pseudocycles were annotated using a Python script that identifies the largest internal cavity bound by the protein after converting the protein to polyalanine and then identifies all side chain residues contacting this internal cavity. The annotated pockets of the randomly selected pseudocycles were checked manually to confirm that annotations were accurate. The scaffold and annotation of NTF2 pocket residues have been reported previously[16]. Previously verified native small-molecule-binding proteins were taken from the PDBBind database[15]. Only single-chain native small-molecule-binding proteins were selected to be comparable with pseudocycles. The binding pockets were selected on the basis of the annotation provided by PDBBind database. The native proteins were relaxed to remove protein side chain clashes with backbone and side chain constraints using Rosetta to remove clashes before any further computational experiments.

Nineteen ligands were used for visual docking and design experiments: xanthurenic acid (4KL), morphine (MOI), lumiflavin (LFN), lenvatinib (LEV), cholic acid (CHD), fentanyl (7v7), dexamethasone (DEX), aspirin (AIN), fosfomycin (FCN), folic acid (FOL), methotrexate (MTX), diltiazem (D6C), dapsone (DSP), levofloxacin (LFX), ascorbic

acid (ASC), JQ1, phosphatidylserine (PS1), pyridoxal (PXL), and phosphatidylinositol 3-phosphate (PI1). The protonation states of all ligands were used when the pH was at 7.4.

The same procedure was used to dock all 19 ligands to pseudocycles, NTF2, and native proteins. One to eight rotamers of each ligand were extracted from the PDB or Cambridge Structure Database[25]. Hydrogens were added to ligand rotamers using OpenBabel[26] or VMD[27] with visual inspection. The conjugation and charge were edited or added with VMD or Chimera[28] with visual inspection. The parameter file of the ligand was generated using the Python script from Rosetta application.

Rifgen/RIFdock suite[13] was used to perform protein–ligand docking. Various amino acid rotamers (referred to as RIFs) that provide hypothetical polar, aromatic, and apolar interactions to the ligand rotamer were generated for each ligand using the Rifgen function, with the requirement of polar interactions to all heavy atoms from the ligand rotamer. This requirement was dropped if the RIF was smaller than 1 MB, because RIFs of this size often do not yield meaningful docking data. The RIFs for each ligand, which encode the geometry and energy information for potential interactions between amino acid rotamers and ligand rotamers, were docked to pseudocycles, NTF2s, and native proteins at their annotated pocket residues using RIFdock. All remaining requirements to make polar interactions to ligand heavy atoms were kept during the docking procedure. For each protein scaffold, a maximum of 30 docks was generated. At this step, for many of the protein scaffolds, the pocket failed to accommodate the ligand rotamer or to provide positions to hold the required interacting amino acid rotamers.

The generated docks were designed using the Rosetta sequence design suite to provide score terms to identify the most-suitable protein scaffolds for holding each ligand. Each generated dock was designed using a fast version of the fix-backbone sequence design procedure[14]. Previous studies have suggested that this version generates interface metrics that are highly correlated with scores generated using the slow version of the procedure, and thus it can be used to design large numbers of docks for selection of promising docks for binder design[14]. Interface metrics, including 'contact_molecular_surface' and 'ddG,' were used to select the top percentile of binders for each ligand[14]. For the top 1% of these docks, scaffold-type origin was identified to determine which scaffold group (Fig. 4b,c, pseudocycle, NTF2, native protein) accommodated the widest array of ligands.

### Expression and purification of selected proteins

All chemicals and supplies were purchased from Thermo Fisher Scientific unless specified otherwise.

Designs were reverse translated into DNA using a custom Python script that attempts to maximize host-specific codon adaptation index[29] and Integrated DNA Technologies synthesizability, which includes optimizing whole-gene and local GC content as well as removing repetitive sequences, and ordered as Eblocks from IDT. Eblocks were cloned into a pET29b-derived vector with carboxy-terminal SNAC-cleavable His tags using Golden Gate assembly (New England Biolabs) and transformed into *E. coli* BL21 strain. The solubility of the proteins was first assessed using small-scale expression. One-milliliter cultures were grown in a round-bottom 96-deep-well plate covered with a breathable film and shaken at 270*g* overnight at room temperature; the cultures were collected by centrifugation for 10 min at 4,000*g* and resuspended in bugbuster lysis buffer (1× bugbuster (Millipore), 25 mM Tris, 100 mM NaCl, pH 8). The lysed cells were spun down, and for each protein, 10 µL of clear supernatant was run on premade 15% SDS–PAGE gel (New England Biolabs) to check for protein in the soluble fraction. Protein bands in the expected molecular range were used to judge protein expression and solubility. Soluble designs were subsequently grown in 50 mL autoinduction

medium in 250 mL baffled Erlenmeyer flasks for assay-scale production (6 h at 37 °C followed by 24 h at 18 °C with shaking at 180 r.p.m. in New Brunswick Innova 44 shakers). Cells for each design culture were collected and resuspended in 30 mL of lysis buffer (25 mM Tris 100 mM NaCl, pH 8, with protease inhibitor tablet) and were lysed sonication (3 min sonication, 10 s pulse, 10 s pause, 60% amplitude). After centrifugation for 30 min at 14,000*g*, soluble fractions were bound to 1 mL Ni-NTA resin (Qiagen) in a Econo-Pac gravity column (BIO-RAD) at 4 °C for 1 h with rotation. The resin was washed with 20 column volumes (CV) of low-salt buffer (50 mM tris, 100 mM NaCl, 50 mM imidazole, pH 8) and with 20 CV high-salt buffer (50 mM tris, 1,000 mM NaCl, 50 mM Imidazole, pH 8).

For initial characterization using SEC (protocol 1) and CD, proteins were eluted with 2 CV of elution buffer (25 mM tris, 100 mM NaCl, 500 mM Imidazole, pH 8) and purified on a superdex 75 increase 10/300 GL column connected to ÄKTA protein purification systems in TBS buffer (25 mM Tris, 100 mM NaCl, pH 8).

For crystallography (protocol 2), the samples were treated as the same as that in protocol 1, except 4 to 8 flasks of 50 mL of culture were pooled together before sonication, and His tags were cleaved on beads (a.k.a. Ni-NTA resin), following the SNAC cleavage protocol[30], before subsequent SEC purification.

For small-angle X-ray scattering (SAXS) studies, the samples were treated as above, except 4 flasks of 50 mL of culture were pooled together before sonication, and His tags were cleavaged on bead, following the SNAC cleavage protocol, before subsequent SEC purification. The sample buffer was exchanged to 20 mM tris, 100 mM NaCl, and 2% glycerol (vol/vol) for SAXS studies.

### Circular dichroism characterization of selected proteins

Circular dichroism spectra were measured with a Jasco J-1500 CD spectrometer. Samples were typically around 0.25 mg mL$^{-1}$ (range 0.1–0.5 mg mL$^{-1}$) in 25 mM phosphate buffer, pH 8, and a cuvette with a path length of 1 mm was used. The CD signal was converted to mean residue ellipticity by dividing the raw spectra by $N \times C \times L \times 10$, where $N$ is the number of residues, $C$ is the concentration of protein, and $L$ is the path length (0.1 cm).

### Crystallographic sample preparation and data analysis

Crystals were produced using the sitting drop vapor diffusion method. Drops with volumes of 200 nL in ratios of 1:1, 2:1, and 1:2 (protein:crystallization) were placed in 96-well plates at 20 °C, using the Mosquito from SPT Labtech. Drops were monitored using the JANSi UVEX imaging system.

For E8, diffraction-quality crystals appeared in a mixture of 0.2 M DL-glutamic acid monohydrate, 0.2 M, DL-alanine, 0.2 M glycine, 0.2 M DL-lysine, 1.0 M imidazole, MES monohydrate (acid), and 37.5% vol/vol of 25% 2-methyl-2,4-pentanediol (MPD; vol/vol) and 25% PEG 1000 and 25% PEG 3350 (wt/vol).

For H10, diffraction-quality crystals appeared in a mixture of 0.12 M D-glucose, 0.12 M D-mannose, 0.12 M D-galactose, 0.12 M L-fucose, 0.12 M D-xylose, 0.12 M *N*-acetyl-D-glucosamine, 0.0499 M HEPES, 0.0501 M MOPS (acid), 20% PEG 500 MME (vol/vol), and 10% PEG 20,000 (wt/vol).

For H12, diffraction-quality crystals appeared in a mixture of 0.09 M sodium fluoride, 0.09 M sodium bromide, 0.09 sodium iodide, 0.0499 M HEPES, 0.0501 M MOPS (acid), 12.5% MPD (vol/vol), 12.5% PEG 1000, and 12.5% PEG 3350 (wt/vol).

Crystals were cryoprotected before being flash frozen in liquid nitrogen before being shipped for data collection at synchrotron. Data collection was performed with synchrotron radiation at the Advanced Photon Source (APS) on beamline 24ID-C.

X-ray intensities and data reduction were evaluated and integrated using either XDS[31] or HKL3000 (ref. 32) and merged and scaled using Pointless and Aimless in the CCP4 program suite[33]. Structure

# Article

determination and refinement starting phases were obtained by molecular replacement using Phaser[34] using the design model for the structures. Following molecular replacement, the models were improved using Phenix autobuild[35]; efforts were made to reduce model bias by setting rebuild-in-place to false and using simulated annealing. Structures were refined in Phenix[35]. Model building was performed using COOT[36]. The final model was evaluated using MolProbity[37]. Data collection and refinement statistics are available in Table 1. Data deposition, atomic coordinates, and structure factors reported in this paper have been deposited in the PDB (8FJE for E8, 8FJF for H10, and 8FJG for H12).

## Reporting summary

Further information on research design is available in the Nature Portfolio Reporting Summary linked to this article.

## Data availability

All data generated in our study have been made freely available. The raw data of SEC and SAXS have been provided. Coordinates and structure factors have been deposited in the Research Collaboratory for Structural Bioinformatics Protein Data Bank with the accession codes 8FJF (H10), 8FJG (H12), and 8FJE (E8). The model of all 9,838 unique scaffolds and the sequences and the design models of the 96 characterized designs have been uploaded to GitHub (https://github.com/LAnAlchemist/Psedocycles_NSMB.git). The biochemical and biophysical characterization of the designs, structure prediction calculations, sequence analysis, and X-ray crystallography statistics are provided as Supplementary Figures and Tables.

## Code availability

The Rosetta macromolecular modeling suite (http://www.rosettacommons.org) is freely available to academic and non-commercial users. The scaffold-generation scripts were uploaded to github (https://github.com/dmitropher/af2_multistate_hallucination.git).

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

## Acknowledgements

This work was supported by a grant from the Department of Defense (DOD-0001039633, D.Z., D.B.), a grant from the National Institute on Aging (5U19AG065156, D.R.H., D.B.), a gift from the Washington Research Foundation (L.A.), a gift from Microsoft (J.D., D.B.), a Human Frontier Science Program Cross Disciplinary Fellowship (LT000395/2020-C, L.F.M.), an EMBO Non-Stipendiary Fellowship (ALTF 1047-2019, L.F.M.), an EMBO long-term fellowship (ALTF 139-2018, B.I.M.W.), the Howard Hughes Medical Institute (A.C., D.B.), the Audacious Project at the Institute for Protein Design (A.K.B., A.K., L.C., D.B.), the Open Philanthropy Project Improving Protein Design Fund (B.I.M.W., H.N., D.B.), and E. and W. Schmidt, and Schmidt Futures funding from E. and W. Schmidt by recommendation of the Schmidt Futures program (L.C., D.B.).

This work is based upon research at the Northeastern Collaborative Access Team beamlines, which are funded by the National Institute of General Medical Sciences from the National Institutes of Health (P30 GM124165). This research used resources of the Advanced Photon Source, a U.S. Department of Energy (DOE) Office of Science User Facility operated for the DOE Office of Science by Argonne National Laboratory under contract no. DE-AC02-06CH11357. We thank M. Baek for her help on using RoseTTAFold. We thank X. Li for her help on verifying protein expression using electrospray ionization high-resolution mass spectroscopy. We thank R. Ragotte for his suggestions on crystallization studies. We thank W. Sheffler for his homogeneous transform package, 'homog.' We thank S. Gerben for her suggestion on plotting pocket characteristics. We thank I. Anishchenko for his help with sequence-similarity comparison with native proteins. We thank R. Kibler for his help with SAXS experiments. We thank F. Praetorius for his editing and suggestions during manuscript preparation. We thank Microsoft and AWS for generous gifts of cloud computing credits.

This work was conducted at the Advanced Light Source (ALS), a national user facility operated by Lawrence Berkeley National Laboratory on behalf of the Department of Energy, Office of Basic Energy Sciences, through the Integrated Diffraction Analysis Technologies (IDAT) program, supported by DOE Office of Biological and Environmental Research. Additional support comes from the National Institute of Health project ALS-ENABLE (P30 GM124169) and a High-End Instrumentation Grant S10OD018483.

This research used resources of the National Energy Research Scientific Computing Center, which is supported by the Office of Science of the US Department of Energy under contract no. DE-AC02-05CH11231.

All first authors contributed equally to this work; author order is decided by alphabetical order of the last names. All co-first authors (L.A., D.R.H., D.Z.) agree that for personal pursuits, the order of their respective names may be changed to best suit their own interests.

## Author contributions

L.A., D.R.H., and D.Z. worked together on the computational pipeline and the molecular biology experiments. J.D. provided the usage of ProteinMPNN and suggestions on sequence design. B.I.M.W., L.F.M., and A.C. wrote the prototype of the MCMC-based scaffold sampling script. A.K.B., H.N., and A.K. performed the crystallography experiments. L.C. performed the sec-mal experiments, and D.B. helped design the project.

## Competing interests

D.B., L.A., D.H., D.Z., J.D., B.W., L.M., and A.C. are the authors of a provisional patent application (63/367,903) submitted by the University of Washington for the design, composition, and function of the proteins created in this study.

## Additional information

**Extended data** is available for this paper at https://doi.org/10.1038/s41594-023-01112-6.

**Correspondence and requests for materials** should be addressed to Linna An or David Baker.

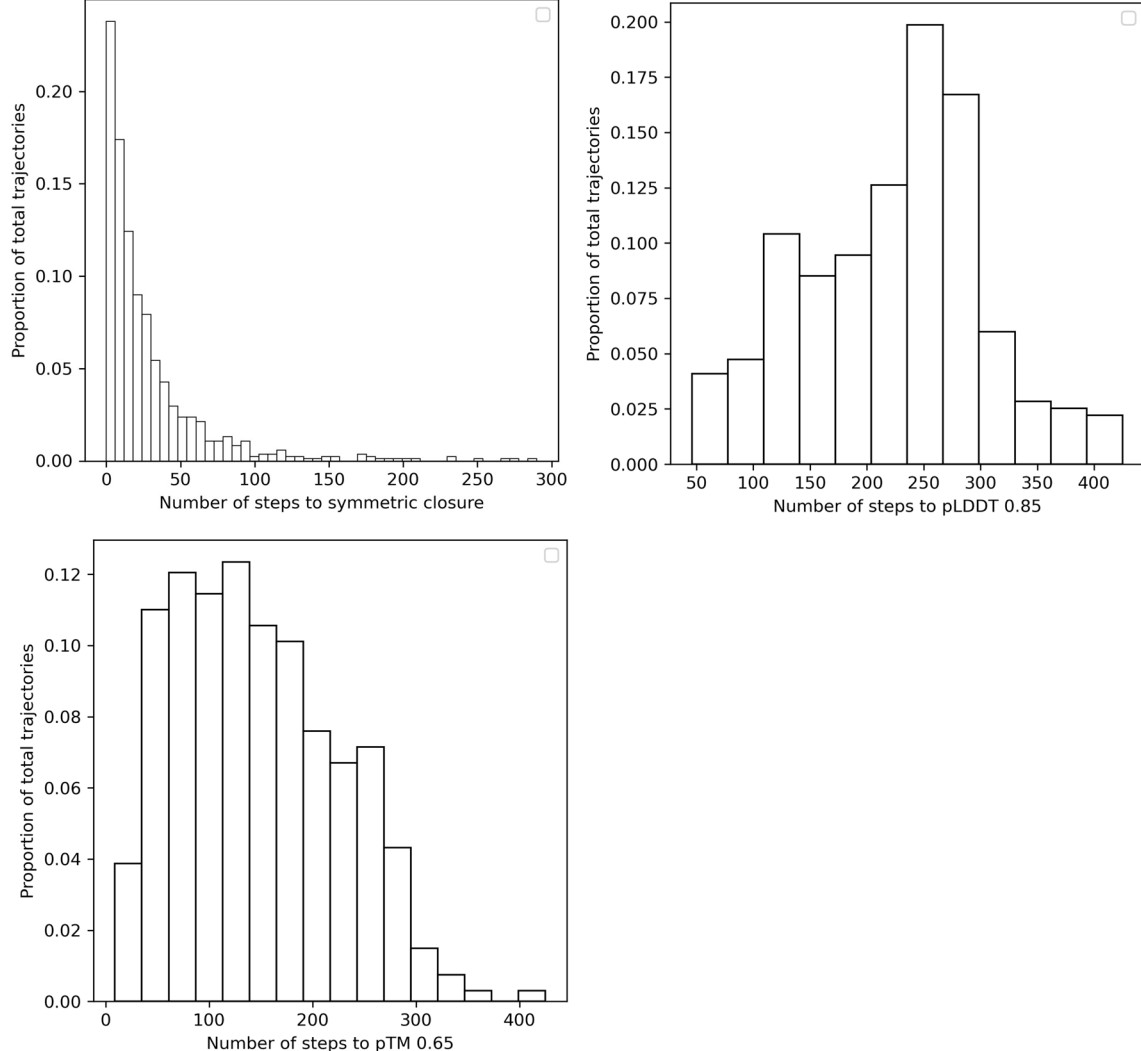

**Extended Data Fig. 1 | The histograms of MCMC steps to convergence.** The number of steps to closure for a representative sample of trajectories is shown here. Symmetric closure is defined as a 'closure score' of 0.1 or less. A clear trend is that AF2 readily predicts closed, cyclic structures from random repetitive sequences, but with very low confidence and quality until a rather large number of mutations.

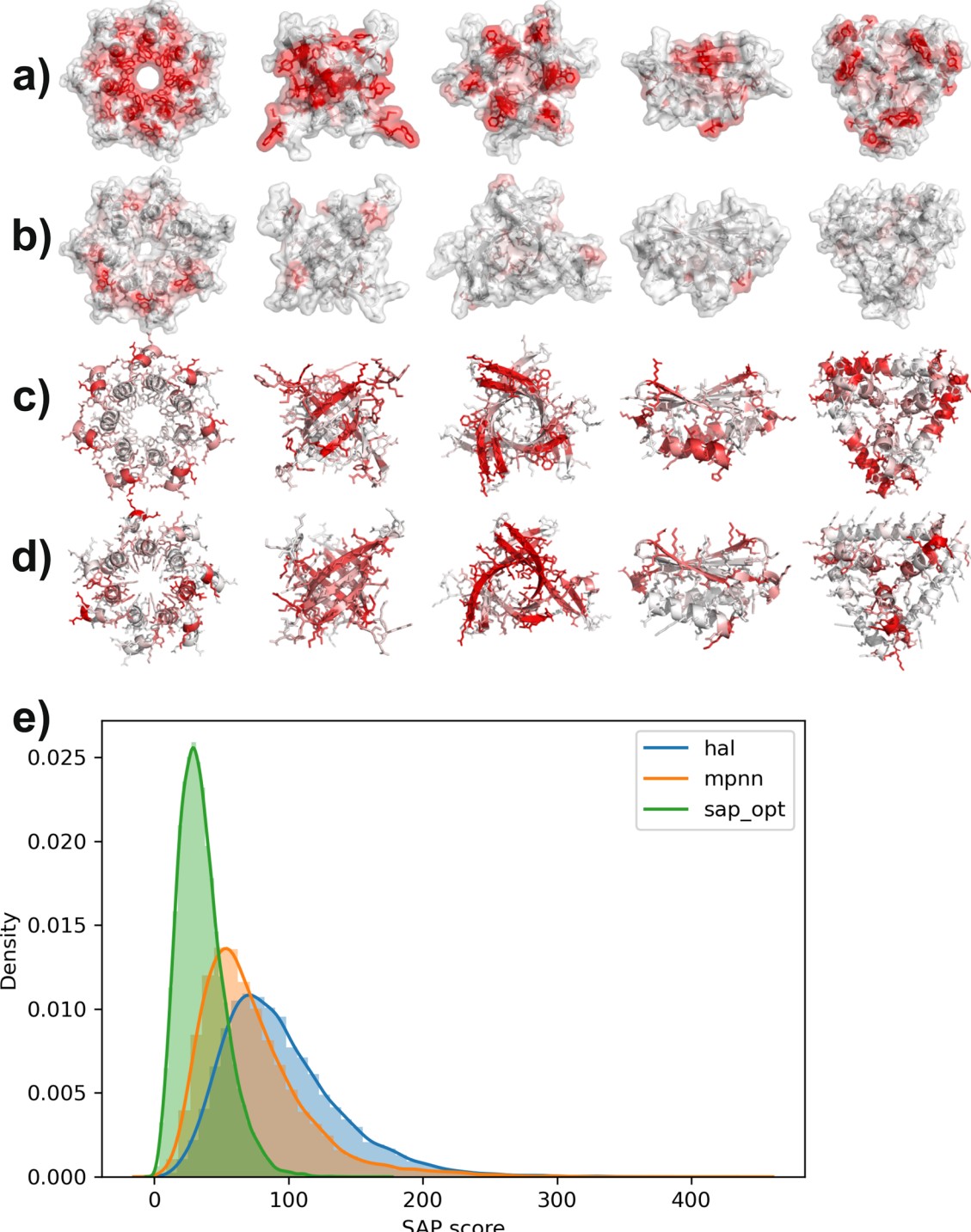

**Extended Data Fig. 2 | Cartoons of exemplary designed pseudocycles from cyclic repeat protein hallucination showing per residue SAP and psipred scores before and after ProteinMPNN+Rosetta redesign.** SAP score was improved during the final design step. **a)** 5 diverse representative proteins following the hallucination procedure colored by SAP score. Color scales from white (no aggregation propensity) to red (high aggregate propensity). **b)** The same 5 proteins after ProteinMPNN redesign and Rosetta surface optimization colored by SAP score. **c)** The same 5 hallucinated proteins colored by agreement of single sequence psipred prediction with the intended secondary structure. Color scales from white (perfect agreement) to red (no agreement). **d)** The redesigned proteins colored by agreement of single sequence psipred prediction with the intended secondary structure. **e)** Histogram of SAP score for original hallucinations (hal), after ProteinMPNN (mpnn) redesign, and after Rosetta surface optimization (sap_opt).

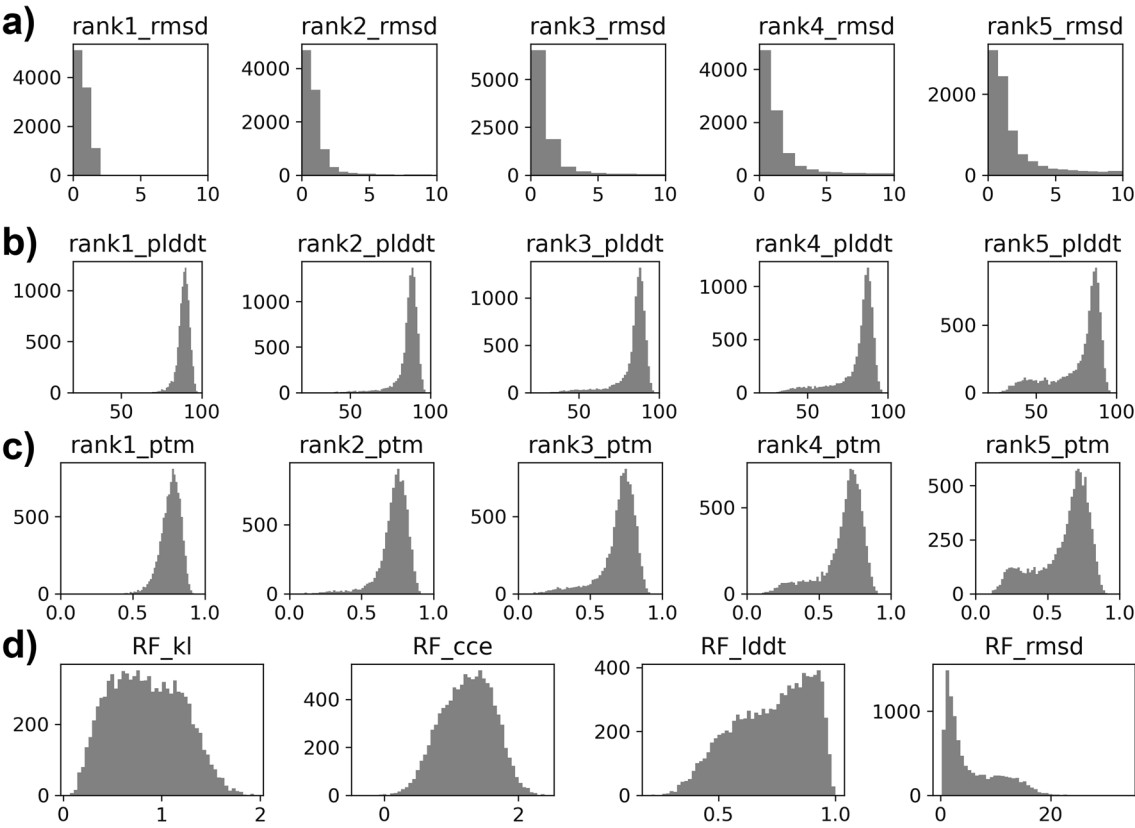

**Extended Data Fig. 3 | AF2 and RF metric histograms for 9838 pseudocycle cluster representatives. a**). AF2 Ca-RMSD to design models for 5 AF2 models by AF2 rank. **b**). AF2 plDDT for predictions. **c**). AF2 ptm for predictions. **d**). RosettaFold (RF) Ca-RMSD to design model, RF lddt, RF KL divergence, and RF CCE for predictions.

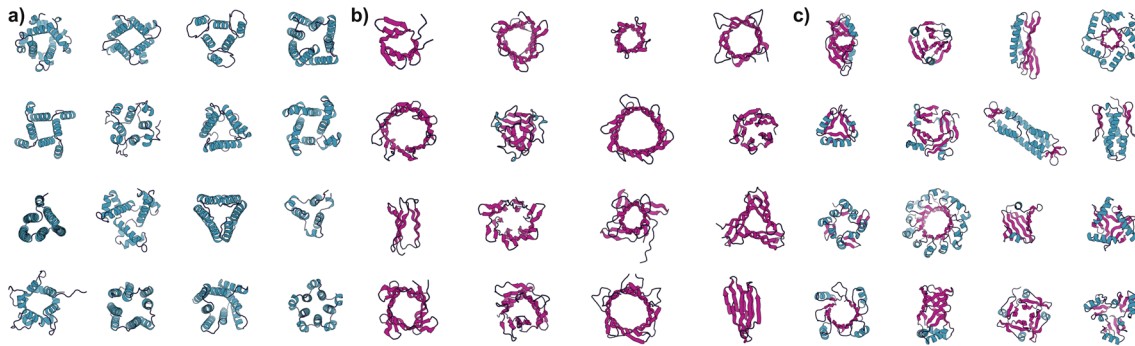

**Extended Data Fig. 4 | Diverse pseudocycle structures.** Exemplary α (**a**), β (**b**), and α/β-containing (**c**) pseudocycles.

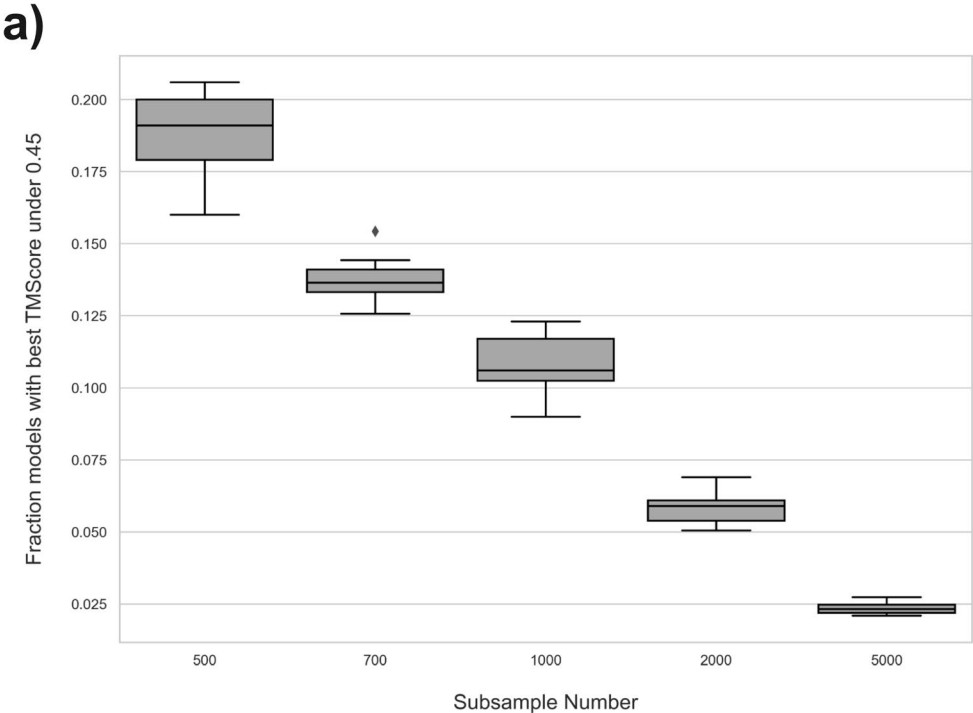

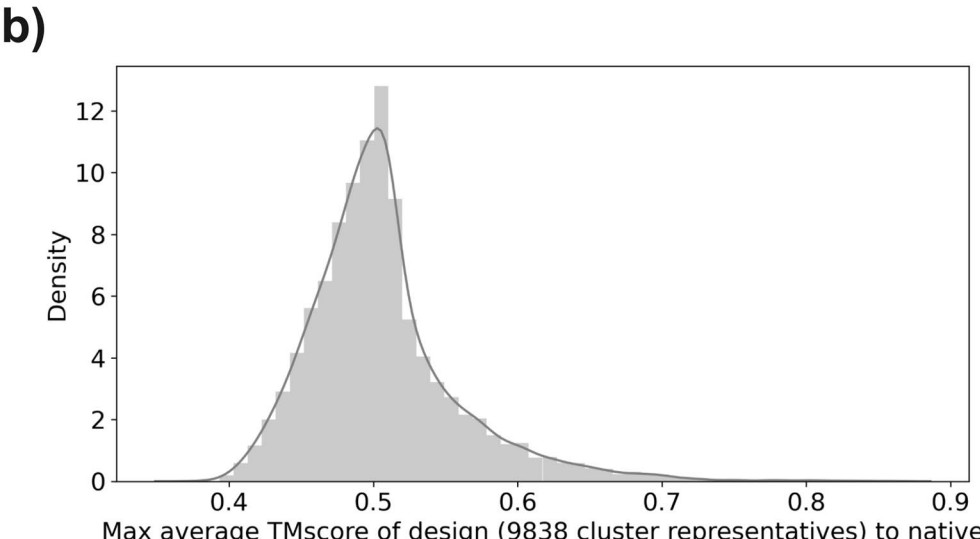

**Extended Data Fig. 5 | Our scaffold generation pipeline samples extensively of pseudocycle structural space, and the TMscore distribution of the designs coming from the pipeline to natives. a**). We randomly constructed 10 subsamples from our pool of design scaffolds (after removing structurally redundant models as described in Methods Protein Clustering) and recorded the number of models we found with TMScore of 0.45 or less to every other model in the pool. This represents models which are significantly different from every other model in the sample. Increased sample size shows that a smaller fraction of the models are structurally unique, this further implies that we have sampled the majority of the space available with this method. Central line shows median, box shows interquartile range, and whiskers show range, except for one outlier for the 700 sample group, shown as a diamond. **b**). For each of our 9838 design cluster representatives, we computed the max average TMscore to native structures and plotted as a histogram.

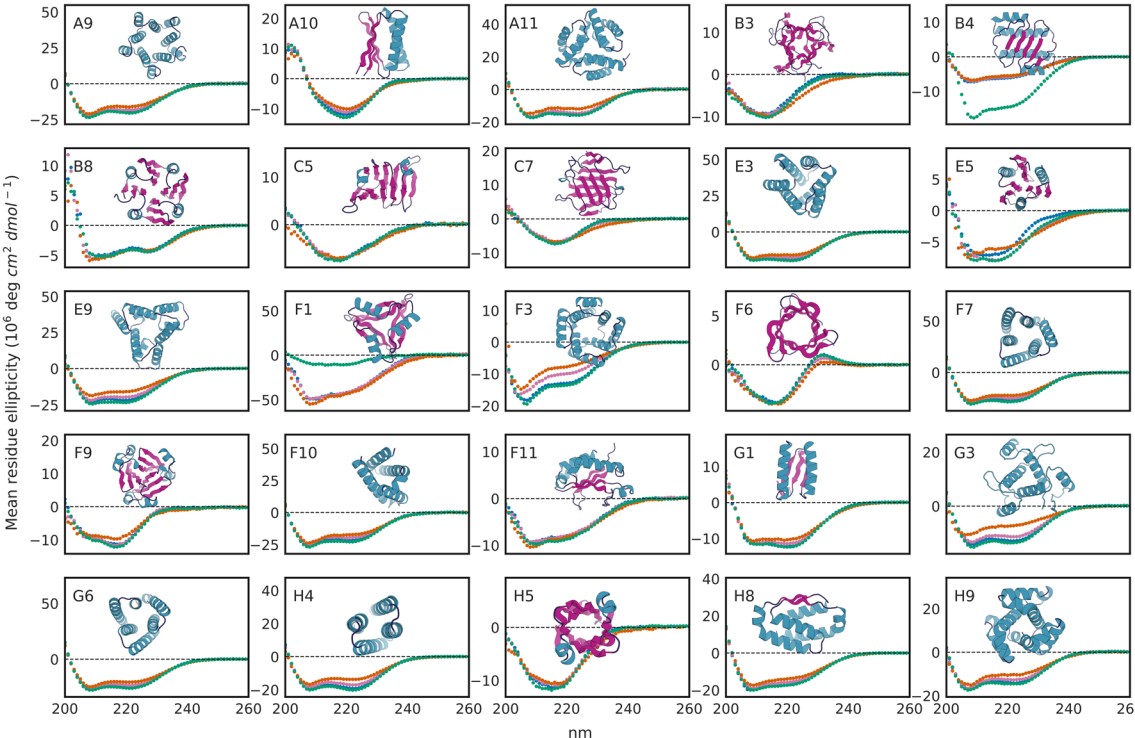

**Extended Data Fig. 6 | CD data for 25 designs not shown in Fig. 2.** Different temperatures of the CD scan spectra are plotted as follows: 25 °C in blue, 55 °C in orange, 95 °C in pink, refolding at 25 °C in green. The cartoon of the corresponding designed pseudocycle is shown with each CD spectra. The sheet, helix, loop substructures are colored in magenta, teal, and dark blue, respectively.

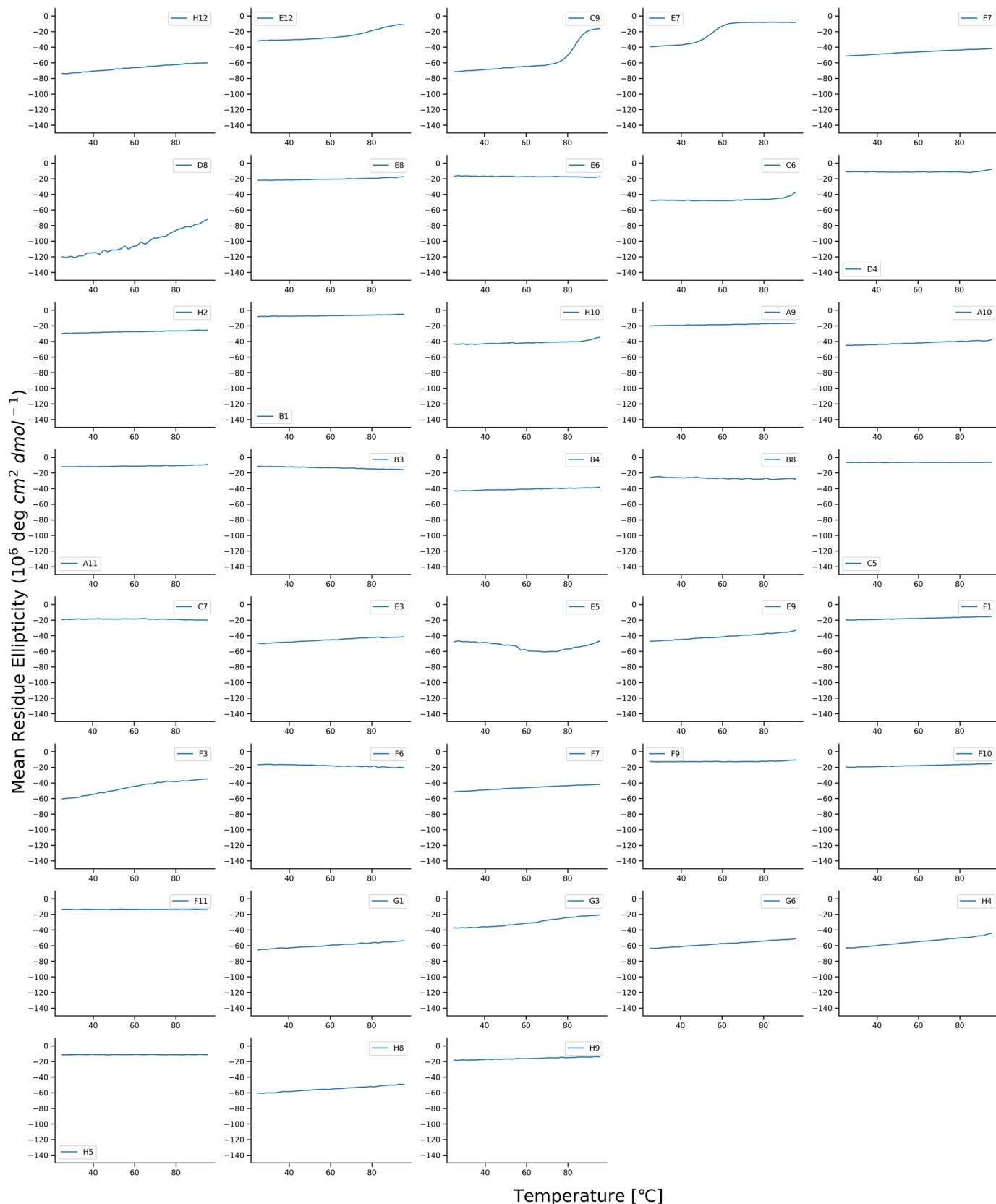

**Extended Data Fig. 7 | The curves of ellipticity as a function of temperature for all designed pseudocycles with monomer fractions judged by SEC traces.** Temperature-dependent CD scan suggested most de novo pseudocycles are highly stable.

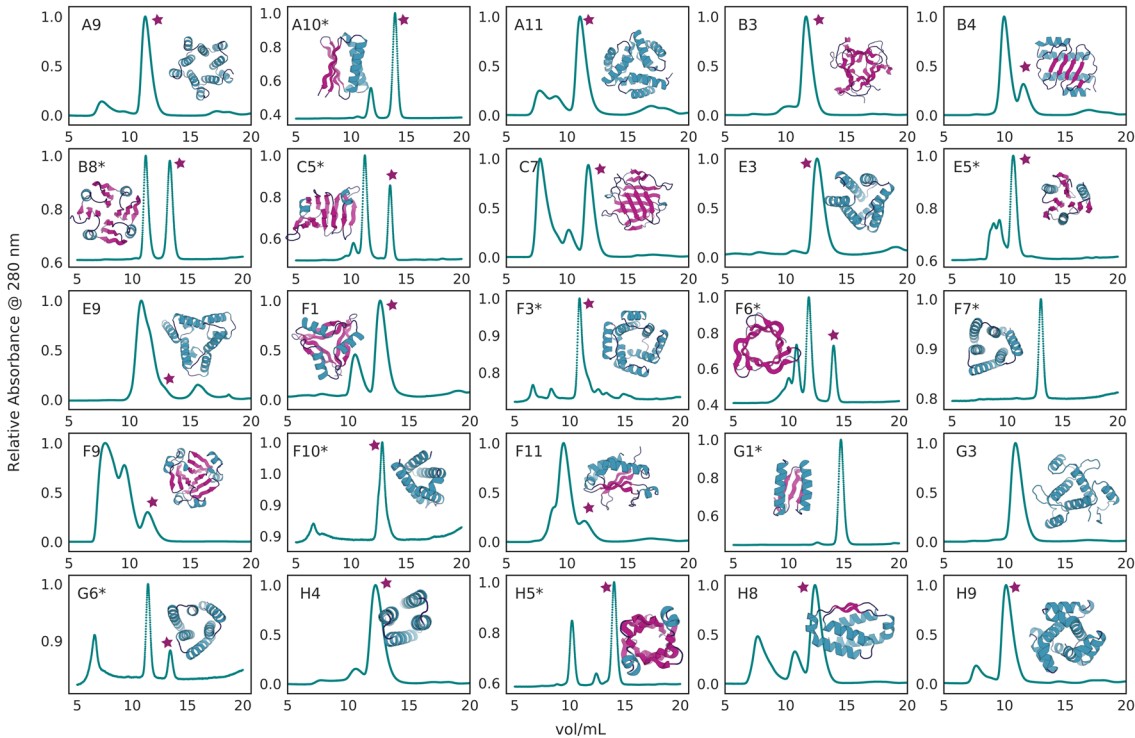

**Extended Data Fig. 8 | SEC data for 25 designs not shown in Fig. 2.** The SEC analysis performed using protocol2 were marked with a star (*) at its label. Monomeric fraction was marked out using a magenta star. The cartoon of the corresponding designed pseudocycle is shown with each subplot. The sheet, helix, loop substructures are colored in magenta, teal, and dark blue, respectively.

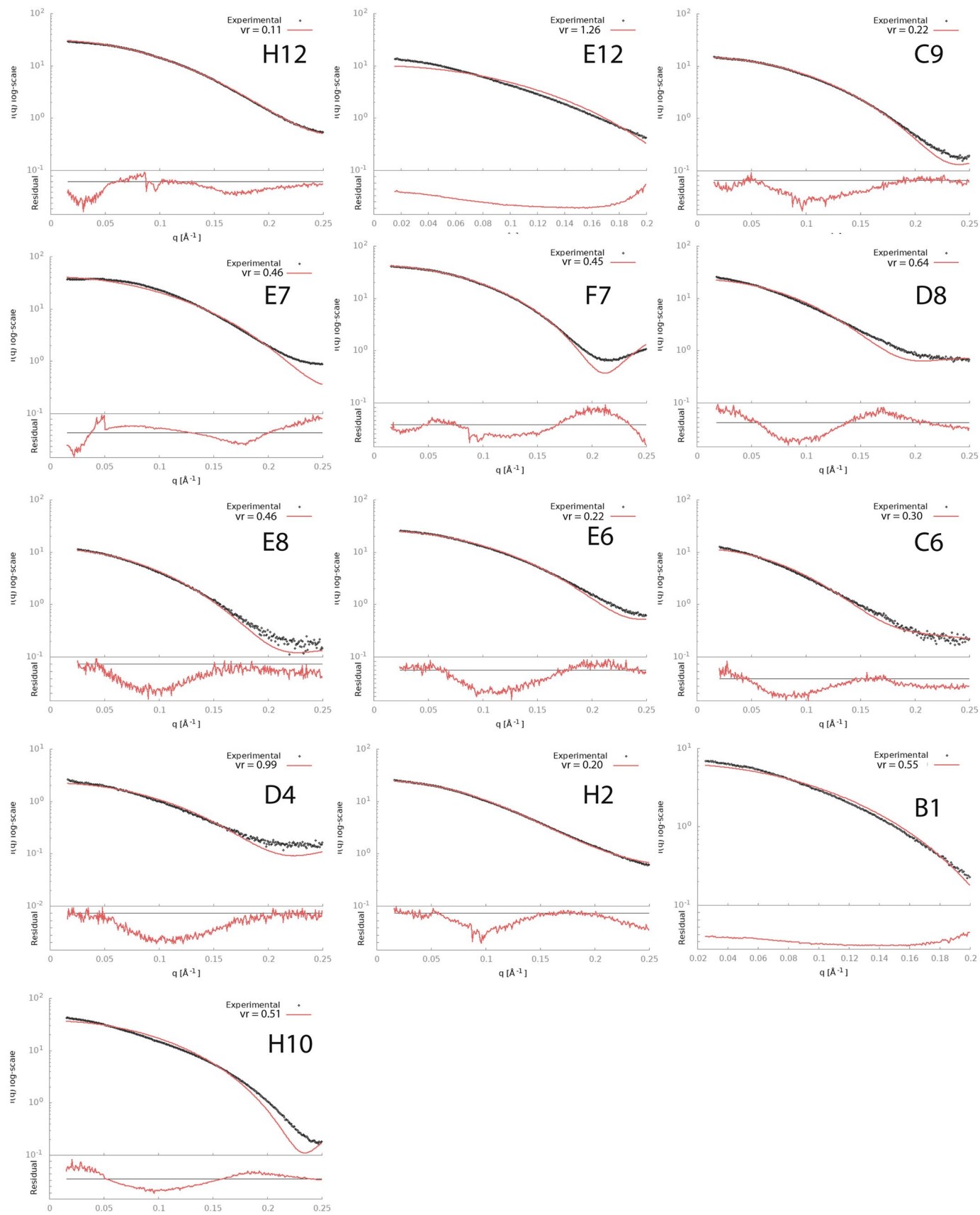

**Extended Data Fig. 9 | Small-angle X-ray scattering of selected pseudocyclic proteins.** The volatility of ratio (vr) is marked and suggests monomeric distribution of all designs.

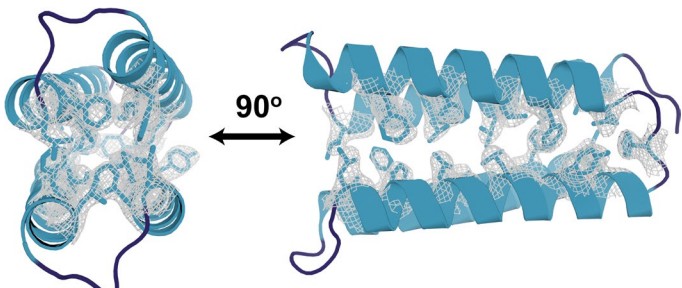

**Extended Data Fig. 10 | The pocket of design H12.** The design of H12 were shown in cartoon and the interface residue side chains were shown in sticks. The electron density of the interface residues were shown in gray mesh.

Corresponding author(s): Linna An*, Derrick R. Hicks*, Dmitri Zorine*, Justas Dauparas, Basile I. M. Wicky, Lukas F. Milles, Alexis Courbet, Asim K. Bera, Hannah Nguyen, Alex Kang, Lauren Carter, and David Baker

# Reporting Summary

## Statistics

For all statistical analyses, confirm that the following items are present in the figure legend, table legend, main text, or Methods section.

| n/a | Confirmed | |
|---|---|---|
| ☐ | ☒ | The exact sample size (*n*) for each experimental group/condition, given as a discrete number and unit of measurement |
| ☐ | ☒ | A statement on whether measurements were taken from distinct samples or whether the same sample was measured repeatedly |
| ☒ | ☐ | The statistical test(s) used AND whether they are one- or two-sided<br>*Only common tests should be described solely by name; describe more complex techniques in the Methods section.* |
| ☐ | ☒ | A description of all covariates tested |
| ☒ | ☐ | A description of any assumptions or corrections, such as tests of normality and adjustment for multiple comparisons |
| ☐ | ☒ | A full description of the statistical parameters including central tendency (e.g. means) or other basic estimates (e.g. regression coefficient) AND variation (e.g. standard deviation) or associated estimates of uncertainty (e.g. confidence intervals) |
| ☒ | ☐ | For null hypothesis testing, the test statistic (e.g. *F*, *t*, *r*) with confidence intervals, effect sizes, degrees of freedom and *P* value noted<br>*Give P values as exact values whenever suitable.* |
| ☒ | ☐ | For Bayesian analysis, information on the choice of priors and Markov chain Monte Carlo settings |
| ☒ | ☐ | For hierarchical and complex designs, identification of the appropriate level for tests and full reporting of outcomes |
| ☒ | ☐ | Estimates of effect sizes (e.g. Cohen's *d*, Pearson's *r*), indicating how they were calculated |

*Our web collection on statistics for biologists contains articles on many of the points above.*

## Software and code

Policy information about availability of computer code

| Data collection | The rosetta macromolecular modeling suite (www.roseuacornmons.org) is freely avaliable to non-commercial users. The model of all unique scaffolds (9838) and the sequences and the design models of the 96 characterized designs were uploaded to github (https://github.com/LAnAlchemist/Psedocycles_NSMB.git). The scaffold generation scripts were uploaded to github (https://github.com/dmitropher/af2_multistate_hallucination.git). |
|---|---|
| Data analysis | The crystallography data were analyzed using COOT. The circular dichroism and size exclusive chromatography data were analyzed using in-house python scripts. |

For manuscripts utilizing custom algorithms or software that are central to the research but not yet described in published literature, software must be made available to editors and reviewers. We strongly encourage code deposition in a community repository (e.g. GitHub). See the Nature Portfolio guidelines for submitting code & software for further information.

## Data

Policy information about availability of data

All manuscripts must include a data availability statement. This statement should provide the following information, where applicable:
- Accession codes, unique identifiers, or web links for publicly available datasets
- A description of any restrictions on data availability
- For clinical datasets or third party data, please ensure that the statement adheres to our policy

All the raw SEC, SAXS data can be provided upon request, all the other types of data have been made available to all. Data deposition, atomic coordinates, and structure factors reported in this paper were deposited in the PDB (PDB ID 8FJE for E8, 8FJF for H10 and 8FJG for H12). The Rosetta macromolecular modeling suite (http://www.rosettacommons.org) is freely available to academic and non-commercial users. The biochemical and biophysical characterization of the designs, structure prediction, sequence analysis, and X-ray crystallography statistics are provided as Supplementary Figures and Tables. The AlphaFold Protein Structure database used for structural analysis is freely available (https://alphafold.ebi.ac.uk).

## Research involving human participants, their data, or biological material

Policy information about studies with human participants or human data. See also policy information about sex, gender (identity/presentation), and sexual orientation and race, ethnicity and racism.

| | |
|---|---|
| Reporting on sex and gender | N/A |
| Reporting on race, ethnicity, or other socially relevant groupings | N/A |
| Population characteristics | N/A |
| Recruitment | N/A |
| Ethics oversight | N/A |

Note that full information on the approval of the study protocol must also be provided in the manuscript.

# Field-specific reporting

Please select the one below that is the best fit for your research. If you are not sure, read the appropriate sections before making your selection.

☒ Life sciences ☐ Behavioural & social sciences ☐ Ecological, evolutionary & environmental sciences

For a reference copy of the document with all sections, see nature.com/documents/nr-reporting-summary-flat.pdf

# Life sciences study design

All studies must disclose on these points even when the disclosure is negative.

| | |
|---|---|
| Sample size | 96 designed proteins were selected for expression, the sample size was determined by the representation of the designed dataset and the availability of the resources. |
| Data exclusions | None of the molecular biology data from the 96 designed proteins were excluded. |
| Replication | The proteins were purified multiple times to get enough material for different experiments such as crystallography or Small-angle X-ray scattering. |
| Randomization | No randomization is needed in any of our molecular biology experiments. |
| Blinding | No blinding is needed in any of our molecular biology experiments. |

# Reporting for specific materials, systems and methods

We require information from authors about some types of materials, experimental systems and methods used in many studies. Here, indicate whether each material, system or method listed is relevant to your study. If you are not sure if a list item applies to your research, read the appropriate section before selecting a response.

## Materials & experimental systems

| n/a | Involved in the study |
|-----|----------------------|
| ☒ | ☐ Antibodies |
| ☒ | ☐ Eukaryotic cell lines |
| ☒ | ☐ Palaeontology and archaeology |
| ☒ | ☐ Animals and other organisms |
| ☒ | ☐ Clinical data |
| ☒ | ☐ Dual use research of concern |
| ☒ | ☐ Plants |

## Methods

| n/a | Involved in the study |
|-----|----------------------|
| ☒ | ☐ ChIP-seq |
| ☒ | ☐ Flow cytometry |
| ☒ | ☐ MRI-based neuroimaging |

