## [Peer Review File · Nature Structural & Molecular Biology]

Peer Review Information

Manuscript Title: Hallucination of closed repeat proteins containing central pockets

Corresponding author name(s): Linna An, David Baker

Reviewer Comments & Decisions:

Decision Letter, initial version:
--

Message: 2nd Nov 2022

Dear Dr. An,

Thank you again for submitting your manuscript "Hallucination of closed repeat proteins containing central pockets". Please accept my sincere apologies for the unusual delay in responding, which resulted from the difficulty in obtaining suitable referee reports. Nevertheless, we now have comments (below) from the 3 reviewers who evaluated your paper. In light of those reports, we remain interested in your study and would like to see your response to the comments of the referees, in the form of a revised manuscript.

You will see that while all reviewers appreciate the application of the hallucination method in designing proteins with central pockets, reviewer #2 asks for additional SAXS data to validate the shapes of the designed scaffolds, and reviewer #3 finds showing experimentally the binding of small molecules to the designed cavities necessary. Editorially, we agree that experimental binding data, at least on a subset of scaffolds, would strengthen the interest and impact of this work. Please be sure to address/respond to all concerns of the referees in full in a point-by-point response and highlight all changes in the revised manuscript text file. If you have comments that are intended for editors only, please include those in a separate cover letter.

Please resubmit your revised manuscript using the link below. We would still consider your revision, provided that no similar work has been accepted for publication at NSMB or published elsewhere.

As you already know, we put great emphasis on ensuring that the methods and statistics reported in our papers are correct and accurate. As such, if there are any changes that

should be reported, please submit an updated version of the Reporting Summary along with your revision.

Reporting Summary:

Please note that all key data shown in the main figures as cropped gels or blots should be presented in uncropped form, with molecular weight markers. These data can be aggregated into a single supplementary figure item. While these data can be displayed in a relatively informal style, they must refer back to the relevant figures. These data should be submitted with the final revision, as source data, prior to acceptance, but you may want to start putting it together at this point.

Data availability: this journal strongly supports public availability of data. All data used in accepted papers should be available via a public data repository, or alternatively, as

Supplementary Information. If data can only be shared on request, please explain why in your Data Availability Statement, and also in the correspondence with your editor. Please note that for some data types, deposition in a public repository is mandatory - more information on our data deposition policies and available repositories can be found below: <https://www.nature.com/nature-research/editorial-policies/reporting-standards#availability-of-data>

Nature Structural & Molecular Biology is committed to improving transparency in authorship. As part of our efforts in this direction, we are now requesting that all authors identified as 'corresponding author' on published papers create and link their Open Researcher and Contributor Identifier (ORCID) with their account on the Manuscript Tracking System (MTS), prior to acceptance. This applies to primary research papers only. ORCID helps the scientific community achieve unambiguous attribution of all scholarly contributions. You can create and link your ORCID from the home page of the MTS by clicking on 'Modify my Springer Nature account'. For more information please visit [visit <http://www.springernature.com/orcid>](http://www.springernature.com/orcid).

[Redacted]

Sincerely,
Sara

Sara Osman, Ph.D.
Associate Editor
Nature Structural & Molecular Biology

Referee expertise:

Referee #1: Drug design, chemical biology

Referee #2: Protein design, structural biology

Referee #3: Protein design, structural biology

Reviewers' Comments:

Reviewer #1:

Remarks to the Author:

In this paper, the authors developed a new and original approach to explore the space of closed repeat proteins. This cyclic procedure generated a large number of novel cyclic backbones. The experimental assays showed that 38 of the 96 selected designs were consistent with the design models. The major motivation for the design of these pseudocycles was to get diverse pockets for ligand binding. Although there is no experimental binding assay data to confirm the ligand binding activity of the designed proteins, using the docking methods in Rosetta suite, they showed that for most small molecule ligand they tested, the most designable binding sites were obtained with the designed pseudocycles. This paper provides a new strategy for de novo design of ligand binding proteins or enzymes.

1) Although 21,021 cyclic protein structures were hallucinated, I am curious about whether it was an exhaustive search or these scaffolds were only a small part of all the possible structures for this defined sequence space. Can the authors show some data to explain this problem?

2) How much time the hallucination procedure took to design one structure?

3) The repeat number may be the most important parameter for the pseudocycle structure. When clustering the designed scaffolds, why the repeat number is not considered first?

4) In the ligand docking studies, the pocket residue annotation for the pseudocycles and the native proteins were not the same. This difference may result in a larger annotated cavity for the pseudocycles? Does this difference affect the comparison of the docking results?

5) It is helpful to provide the structure models for the novel scaffolds, which have a small TMscore.

Reviewer #2:

Remarks to the Author:

The manuscript by An et al illustrates a new design pipeline for closed repeat proteins containing central pockets. The protocol is based on the hallucination process described by the same group. The designs were experimentally validated, including also two crystal structures, and the models and sequences were compared to available proteins present in

the uniref and pdb databases, to highlight the novelty of the sequences and folds. Finally, to evaluate the potential of these designs as scaffolds for small molecule binding, docking and design were performed and assessed for a number of ligands.

The manuscript is original and significant: it expands and combine previous work on design methods and novel repeat proteins, while aiming specifically at designing novel scaffolds for small molecule binding.

The text was clear without mistakes or typos. The methods used were appropriate and well described, with high quality data. Conclusions were appropriate considering the data, but could benefit from further data (see below). References are adequate.

I recommend to accept this manuscript in nsmb, although I suggest to provide clarifications and additional information, as indicated below.

Design method:

Why 1-3 mutation per step of the MCMC protocol? How has this number been chosen?
Why the sequence symmetry was not maintained? What was the rational for this choice?
Could protein mpnn be used while maintaining sequence symmetry?

Additional data

Experimental Characterization of Selected Pseudocycles:

it would be important to see at least some low resolution structural information on the set of 17 folded and mono dispersed proteins. SAXS should be used, as it has been done in other similar papers, to confirm that the shape corresponds to the expected from the design, in particular to demonstrate that the designs are indeed closed circles.

Small molecule binding data, if available, would strengthen the paper, but a full scale binder analysis would probably be outside of the scope of this manuscript. These are desirable, but not a necessary addition to the paper.

Discussion:

Although the author mentioned the advantage of the design protocol compared to previous method, it would be good to have comments on the success rate compared to a previous methods on repeat protein design from Brunette et al (ref 3) and on the speed of the design protocol, with an estimate of the computing time required for an average design or a few designs.^{[1][2]}

Minor corrections

Fig S4 d panel is written as s panel instead

Update references to bioarxiv manuscripts that have been published after peer review, e.g. reference 9

Data availability:

The computational design pipeline should be made available via links to accessible repositories and/or additional files to allow reproducibility and use by the scientific community.

Sequences and models of the 96 tested pseudo cycles (with indication of the successful ones) should be included in the supplementary material.

Reviewer #3:

Remarks to the Author:

The manuscript describes a deep-learning based design approach for cyclic closed repeat proteins that uses as input only the number and length of the repeat. The approach is based on an adapted hallucination procedure that here is coupled to design with ProteinMPNN and Rosetta. The hallucination procedure uses an optimization process in which sequence folding to a cyclic structure is assessed using AlphaFold2 and is repeated until convergence of pLDDT and symmetry loss score. Diverse sets of closed repeat proteins were designed and 96 tested experimentally for expressability and stability. In addition, docking was performed to sample cavities for possible ligand binding design. The paper is an extension of the earlier hallucination paper that incorporates a few new developments and that highlights the use of these newly designed proteins as starting points for ligand binding design simply based on cavities that can be found within them. This is in fact one of the biggest disappointments I had with this manuscript that the potential of these scaffold is speculated but not followed up on. Why was ligand binding not tested experimentally? In fact, did the authors expect cavities and why?

Altogether the entire paper seems rushed and the beginning of the manuscript does not resonate with the discussion. The abstract and introduction start with TIM- and beta-barrel, but these are not mentioned again later. Were none such structures build? While it is understandable that the authors want to show new scaffolds, it is a pity that the comparison stops at the intro stage. The authors further say that the new procedure they describe here is superior to the old one. I am missing a comparison as well as details on the methods. Also going to the original cited papers by some of the same authors did not list enough details to repeat the computation.

Further, how do AlphaFold2 and RoseTTAafold compare. Why were there 5 models with AF2 done but only one with RoseTTAafold?

Experimentally, it is good to see that many proteins can be expressed – however the authors should be careful in their description. Reading the methods section, it appears that in the first step they test protein expression, but not whether the proteins are found in the soluble fraction of the cell extract. If this is true it should be described as 'can be expressed' and not as 'soluble'. This would also explain why less proteins can be purified. I appreciate that the SEC traces are shown, a comparison of expected versus apparent molecular weight would be desirable though as to judge oligomerization states. The authors speak of monodispersity in samples, was light scattering used to test this?

The solution of two crystal structures is certainly nice, even if a four-helix bundle with 2-fold symmetry is not the most exciting example one might expect. With the large cavity between the two α -hairpin it would be nice to show also the contacts that contribute to the seemingly high stability of H12. I write seemingly as there are details missing on the equilibration before taking the CD spectra at different temperatures. To be able to view a T-melt trace would be nice and might be more informative. This is especially interesting as the authors argue that native proteins are not stable enough to introduce new binding or active sites. But how stable are the new proteins in comparison?

The docking results are also a bit confusing. In figure 4a for LFN there is a docking with the pseudocycle structure, but in b and c only docking results with natives and NFT2 are listed.

Overall, this manuscript is not ready for publication and the claims are unfortunately not supported sufficiently.

Author Rebuttal to Initial comments

Response to reviewers

We thank the reviewers for their feedback. In the responses below, the original reviewers' comments are italicized and in bold. The responses are not italicized and follow each question. Edits of the manuscript are italicized, and the page number and the line number are provided where relevant. The revised sentences are colored yellow in the revised version of the manuscript.

Referee #1: Drug design, chemical biology

Referee #2: Protein design, structural biology

Referee #3: Protein design, structural biology

Reviewers' Comments:

Reviewer #1:

Remarks to the Author:

In this paper, the authors developed a new and original approach to explore the space of closed repeat proteins. This cyclic procedure generated a large number of novel cyclic backbones. The experimental assays showed that 38 of the 96 selected designs were consistent with the design models. The major motivation for the design of these pseudocycles was to get diverse pockets for ligand binding. Although there is no experimental binding assay data to confirm the ligand binding activity of the designed proteins, using the docking methods in Rosetta suite, they showed that for most small molecule ligand they tested, the most designable binding sites were obtained with the designed pseudocycles. This paper provides a new strategy for de novo design of ligand binding proteins or enzymes.

We thank the reviewer for this excellent summary and positive assessment of our work. We addressed all the comments below. Additionally, we added one more crystal structure of the designed pseudocycles, which further supported our conclusions that our pipeline can generate de novo cyclic proteins with pockets inside with high accuracy. We added this structural data and related analysis to the manuscript.

1) Although 21,021 cyclic protein structures were hallucinated, I am curious about whether it was an exhaustive search or these scaffolds were only a small part of all the possible structures for this defined sequence space. Can the authors show some data to explain this problem?

This is an excellent question, and we did additional calculations in the revised manuscript to address this point (Figure S6). When we randomly select 500, 700, 1000, 2000, and 5,000 scaffold clusters from the 9838 scaffold clusters that we generated, and look at their all-by-all TMscores, the percentage of the structures with TMscores lower than 0.45 dropped below 2.5%. This means, we have fairly exhaustively covered the spaces since the appearance of totally new scaffolds become low.

Page 3, line 9, new version:

To evaluate how thoroughly our calculations sample the space of possible pseudocycles, we first reduced the structural redundancy (see Methods protein clustering), then we randomly selected 10 subsets of designs with 500, 700, 1000, and 5000 members from the redundancy reduced pseudocycle sets, and for each subset, determined the fraction of designs with structures very different from any other member of the subset (TMscore lower than 0.45). While for the smallest subsample sizes, the fraction of singleton scaffolds approached 20%, with increasing subsample sizes this fell to below 2.5% (supplementary Figure. S6). Thus our structure generation by hallucination procedure has identified almost all pseudocycle solutions that pass our selection criteria multiple times, suggesting that our set of 21,021 designs fairly comprehensively covers the space of pseudocycles that can be generated using our approach.

2)How much time the hallucination procedure took to design one structure?

The hallucination trajectories took 20 to 50 minutes on a single GPU, depending on the number of steps and protein length.

3) The repeat number may be the most important parameter for the pseudocycle structure. When clustering the designed scaffolds, why the repeat number is not considered first?

During scaffold clustering, we did first group all the 21,021 scaffolds into different groups based on the initial symmetry number (2 - 7), then grouped all scaffolds based on their repeat unit DSSP, followed by clustering using all-by-all TAlign score. We apologize for the lack of clarity in the original draft, the supplementary information has been edited to more clearly convey this information.

PAGE 12, line 40, original:

Structural clusters were chosen through a combination of hierarchical subdivision by DSSP

character, followed by AgglomerativeCluster (from scikitlearn²²) on an all-by-all matrix of TMalign scores.

PAGE 12, line 40, new version:

All scaffolds were first grouped based on their initial symmetry number (2-7), then grouped based on their repeat unit DSSP. Structural clusters were chosen through a combination of hierarchical subdivision by DSSP character, followed by AgglomerativeCluster (from scikitlearn²²) on an all-by-all matrix of TMalign scores within each group.

4) In the ligand docking studies, the pocket residue annotation for the pseudocycles and the native proteins were not the same. This difference may result in a larger annotated cavity for the pseudocycles? Does this difference affect the comparison of the docking results?

The pocket annotation of native proteins was based on native binding site. The pocket annotation of the pseudocycles was using a rolling ball-based script and double checked manually. Both pocket annotation steps emphasize on the correct recognition of the pocket locations rather than the sizes of the pockets to guarantee successful docking in the following procedure.

More generally, the size of the pockets is likely not the key factor for binding success, instead, the shape complementarity is generally the deciding factor for the success of the virtual docking experiments. We chose ligands with different size and shapes for the virtual docking experiments to eliminate the bias from the pocket size. Based on the virtual docking results, ligands with small sizes, such as xanthurenic acid (4KL) and fosfomycin (FCN), and ligand with larger sizes, such as phosphatidylinositol 3-phosphate (PI1), are well fit to pockets with better binding scores from the pool of pseudocyclic scaffolds than the pool of natives or previous design pools (Figure 4). The final docking score is not based upon the specific geometry of the pocket, but rather its complementarity to the ligand in the docked conformation.

PAGE 13, line 8, added:

The annotated pockets of the randomly selected pseudocycles were checked manually to guarantee annotation accuracy.

5) It is helpful to provide the structure models for the novel scaffolds, which have a small TMscore.

The model of all unique scaffolds (9838) and the sequences and the design models of the 96 characterized pseudocyclic proteins were uploaded to github (https://github.com/LAnAlchemist/Pseudocycles_NSMB.git), which will be open to the public after the publication of the manuscript.

Page 12, line 17:

The model of all unique scaffolds (9838) and the sequences and the design models of the 96 characterized designs were uploaded to github (https://github.com/LAnAlchemist/Psedocycles_NSMB.git).

Reviewer #2:

Remarks to the Author:

The manuscript by An et al illustrates a new design pipeline for closed repeat proteins containing central pockets. The protocol is based on the hallucination process described by the same group. The designs were experimentally validated, including also two crystal structures, and the models and sequences were compared to available proteins present in the uniref and pdb databases, to highlight the novelty of the sequences and folds. Finally, to evaluate the potential of these designs as scaffolds for small molecule binding, docking and design were performed and assessed for a number of ligands.

The manuscript is original and significant: it expands and combine previous work on design methods and novel repeat proteins, while aiming specifically at designing novel scaffolds for small molecule binding.

The text was clear without mistakes or typos. The methods used were appropriate and well described, with high quality data. Conclusions were appropriate considering the data, but could benefit from further data (see below). References are adequate.

I recommend to accept this manuscript in nsmb, although I suggest to provide clarifications and additional information, as indicated below.

We thank the reviewer for this excellent summary and positive assessment of our work. We addressed all the comments below. Additionally, we added one more crystal structure of the designed pseudocycles, which further supported our conclusions that our pipeline can generate de novo cyclic proteins with pockets inside with high accuracy. We added this structural data and related analysis to the manuscript.

Design method:

Why 1-3 mutation per step of the MCMC protocol? How has this number been chosen?

We investigated a variety of mutation schedules early on as part of parameter optimization for the search, single mutation, 3-1, 5-1, 7-1 and others. The mutation schedule did not significantly affect model quality, trajectory duration to convergence, or model diversity. The 3-1 schedule (3 mutations per step early on and 1 mutations per step towards the end) was chosen based on the intuition that sequence should improve better with smaller steps after a satisfactory neighborhood of structure space was found.

Why the sequence symmetry was not maintained? What was the rational for this choice? Could protein mpnn be used while maintaining sequence symmetry?

The pseudocycles were created for generating proteins with pockets for small molecule ligand binding. Considering most of the small molecule ligands of interest are not symmetric, the symmetry of the binding pockets were not maintained. The published version of ProteinMPNN does allow maintenance of the sequence symmetry. We clarified this idea in the revised manuscript to make it clearer to the readers.

Page 2, line 35:

As the backbones are intended for ligand binder design, and most ligands are not symmetric, we used ProteinMPNN to design new sequences given the hallucinated backbones (see Online Methods Protein generation and sequence design pipeline) without requiring sequence repeat symmetry, which resulted in sequence-asymmetric final designs (Figure. 1c).

Additional data

Experimental Characterization of Selected Pseudocycles:

it would be important to see at least some low resolution structural information on the set of 17 folded and mono dispersed proteins. SAXS should be used, as it has been done in other similar papers, to confirm that the shape corresponds to the expected from the design, in particular to demonstrate that the designs are indeed closed circles.

We agree with the comments from the reviewer and will add the SAXS data for the 13 folded monomeric proteins from Figure 2, as a supplementary figure (**supplementary Figure. S11**).

Small molecule binding data, if available, would strengthen the paper, but a full scale binder analysis would probably be outside of the scope of this manuscript. These are desirable, but not a necessary addition to the paper.

We have addressed this point to the editors previously.

Discussion:

Although the author mentioned the advantage of the design protocol compared to previous method, it would be good to have comments on the success rate compared to a previous methods on repeat protein design from Brunette et al (ref 3) and on the speed of the design protocol, with an estimate of the computing time required for an average design or a few designs.

This is an excellent question, and we will answer it at both the computational and experimental level in the revised manuscript.

At the computational level, the success rate depends not only on the time required to generate a single design, but also on how designs are filtered prior to ordering, and what fraction of designs pass the filters. For the Rosetta-based approach, the success rate historically has been heavily linked to the scaffold type (for example, helical proteins usually have significantly higher success rate to be generated than β sheet-rich proteins). For the helical repeat proteins designed in Brunette et al, the reported scaffold design success rate is $66,776/2,880,000 = 2.3\%$, and the sequence design success rate is $11,243/66,776 = 16.8\%$ based on Rosetta scores. The de novo pseudocyclic proteins have much higher scaffold quality. 21,021 out of roughly 28,000 generated pseudocycles (73.84%) were successfully designed with promising scores from Rosetta, AF2, and RF. We will include these data in the revised manuscript.

In terms of compute time used, Rosetta-based approach requires relatively long computational time for both scaffold generation and sequence design steps. The actual computation time highly depends on the size and the fold type of the designed protein. For example, for NTF2-like proteins which are around 120 aa long and contain both helix and β -sheets, the scaffold generation time may take as long as 20 min CPU time and the sequence design step may take 10-30 min CPU time depending on the residue design space (Basanta et al, PNAS 2020). In comparison, our mean number of steps to convergence was roughly 100 steps for successful trajectories, and 500 steps (the maximum steps allowed by our parameterization) for failed runs. The runtime of a 500 step trajectory varies somewhat based on model size and sequence, but we found that under 50 minutes was normal, with some trajectories completing in under 30 minutes.

At the experimental level, the soluble expression level is much higher with the new protocol across all types of protein folds. Similar to computational success rate, the success rate of the scaffolds generated through Rosetta-based protocols is usually highly correlated with the protein fold type. The helical proteins usually have significantly higher solubility rate than β -sheet containing proteins. The helical repeat proteins can have success solubility rate up to $74/83 = 89.2\%$ while the success rate for NTF2-like proteins was $17/64 = 26.6\%$ (Basanta et al, PNAS 2020), while for the new ML protocol, it was 84.4% (81/96), regardless of topology. Their solubility and folding rate are significantly higher than previous work as described in our manuscript.

The most important difference between the old and new protocol is, of course, that the new protocol can freely sample structure space and generate highly diverse folds with different secondary structure content, etc, whereas in the original protocol the secondary structure composition had to be precisely specified.

Page 4, line 42:

Compared to previous Rosetta-based scaffold generation pipelines, our pipeline can freely sample structure space and generate widely diverse protein architectures. Our pipeline also has significantly higher efficiency at both the computational and experimental levels. 21,021 out of roughly 28,000 (73.84%) generated pseudocycles had Rosetta, AF2, and RF metrics predictive

of folding. By comparison, for the simpler problem of helical repeat protein design, the success rate for a previous Rosetta-based protocol was only $11,243/2,880,000 = 0.39\%$. At the experimental level, 84.4% (8196) of the designs from our pipeline are highly soluble, compared to $17/64 = 26.6\%$ for previous Rosetta based design of small molecule-binding (NTF2) scaffolds 16 (the solubility rate for Rosetta-based design of helical repeat proteins lacking cavities is $74/\beta$ repeats that close to 83 = 89.2%).

Minor corrections

Fig S4 d panel is written as s panel instead

The label of Figure S4 was corrected.

Update references to bioarxiv manuscripts that have been published after peer review, e.g. reference 9

The references were updated.

Data availability:

The computational design pipeline should be made available via links to accessible repositories and/or additional files to allow reproducibility and use by the scientific community.

The scripts for generating the pseudocyclic proteins were published online at github (https://github.com/dmitropher/af2_multistate_hallucination.git).

Page 12, line 21:

The scaffold generation scripts were uploaded to github (https://github.com/dmitropher/af2_multistate_hallucination.git).

Sequences and models of the 96 tested pseudo cycles (with indication of the successful ones) should be included in the supplementary material.

The sequences and the design models of the 96 characterized pseudocyclic proteins were uploaded to github (https://github.com/LAnAlchemist/Psedocycles_NSMB.git), which will be open to the public after the publication of the manuscript.

Page 12, line 17:

The model of all unique scaffolds (9838) and the sequence and the design model of the 96

characterized designs were uploaded to github
(https://github.com/LAnAlchemist/Psedocycles_NSMB.git).

Reviewer #3:

Remarks to the Author:

The manuscript describes a deep-learning based design approach for cyclic closed repeat proteins that uses as input only the number and length of the repeat. The approach is based on an adapted hallucination procedure that here is coupled to design with ProteinMPNN and Rosetta. The hallucination procedure uses an optimization process in which sequence folding to a cyclic structure is assessed using AlphaFold2 and is repeated until convergence of pLDDT and symmetry loss score. Diverse sets of closed repeat proteins were designed and 96 tested experimentally for expressability and stability. In addition, docking was performed to sample cavities for possible ligand binding design. The paper is an extension of the earlier hallucination paper that incorporates a few new developments and that highlights the use of these newly designed proteins as starting points for ligand binding design simply based on cavities that can be found within them. This is in fact one of the biggest disappointments I had with this manuscript that the potential of these scaffold is speculated but not followed up on.

Why was ligand binding not tested experimentally?

While designing de novo protein for small molecule ligand binding is the ultimate aim for the grand challenge of small molecule binder design, the extensive binder designs and verification steps are out of the scope of current manuscript. The grand challenging of small molecule binder design includes several big problems, 1) the generation of high quality diverse pocket-containing proteins; 2) the generation of protein-small molecule binder pairs with high shape complementarity; 3) the careful design of the protein-small molecule binding interface for binding affinity. Each of these individual steps are required to solve the grand challenge of the small molecule binder design problem. The current manuscript focuses on solving the first problem, which was not as readily addressed before the development of the deep learning tools published in 2021. Extensive pipeline development and molecular biological verification experiments along with x-ray crystallography will be required to generate and rigorously assess binder designs experimentally.

In fact, did the authors expect cavities and why?

We did, and we modified the manuscript to explain the reason more clearly in the revised manuscript. Cyclic proteins, by definition, have a repeating fold surrounding a central axis.

Because of steric exclusion, no structural element can be directly on the axis as it would then clash with symmetry mates from other repeats (which are at the same distance from this axis). In some cases, there is only a small pore at the axis, while in other more barrel-like structures there is a large cavity in the center.

Page 2, line 25:

Because of steric exclusion of the closed cyclic structures, individual structural elements avoided clashing with the symmetry axis, and formed cavities of various sizes in the center.

Altogether the entire paper seems rushed and the beginning of the manuscript does not resonate with the discussion. The abstract and introduction start with TIM- and beta-barrel, but these are not mentioned again later. Were none such structures build?

We tried to address the stylistic point in the revision, but we noted that the first two reviewers felt quite the opposite.

Our procedure did generate TIM barrels and β barrels, and we changed supplementary figure S5 to showcase some of these structures (**supplementary Figure S5 a, b**). We thank the reviewer for noting this point.

While it is understandable that the authors want to show new scaffolds, it is a pity that the comparison stops at the intro stage. The authors further say that the new procedure they describe here is superior to the old one. I am missing a comparison as well as details on the methods. Also going to the original cited papers by some of the same authors did not list enough details to repeat the computation.

We included a link to a github containing code for reproduction (https://github.com/dmitropher/af2_multistate_hallucination.git). Additionally, we also provided the structure of the 96 characterized pseudocycles and all 9838 unique pseudocycle structures (https://github.com/LAnAlchemist/Pseudocycles_NSMB.git). Both of the git repositories will be open to the public after manuscript publication.

The comparison between the old pipeline and the new pipeline is an excellent question, and we will answer it at both the computational and experimental level in the revised manuscript.

At the computational level, the success rate depends not only on the time required to generate a single design, but also on how designs are filtered prior to ordering, and what fraction of designs pass the filters. For the Rosetta-based approach, the success rate historically has been heavily linked to the scaffold type (for example, helical proteins usually have significantly higher success rate to be generated than β sheet-rich proteins). For the helical repeat proteins designed in Brunette et al, the reported scaffold design success rate is $66,776/2,880,000 = 2.3\%$, and the sequence design success rate is $11,243/66,776 = 16.8\%$ based on Rosetta

scores. The de novo pseudocyclic proteins have much higher scaffold quality. 21,021 out of roughly 28,000 generated pseudocycles (73.84%) were successfully designed with promising scores from Rosetta, AF2, and RF. We included these data in the revised manuscript.

In terms of compute time used, Rosetta-based approach requires relatively long computational time for both scaffold generation and sequence design steps. The actual computation time highly depends on the size and the fold type of the designed protein. For example, for NTF2-like proteins which are around 120 aa long and contain both helix and β -sheets, the scaffold generation time may take as long as 20 min CPU time and the sequence design step may take 10-30 min CPU time depending on the residue design space (Basanta et al, PNAS 2020). In comparison, our mean number of steps to convergence was roughly 100 steps for successful trajectories, and 500 steps (the maximum steps allowed by our parameterization) for failed runs. The runtime of a 500 step trajectory varies somewhat based on model size and sequence, but we found that under 50 minutes was normal, with some trajectories completing in under 30 minutes.

At the experimental level, the soluble expression level is much higher with the new protocol across all types of protein folds. Similar to computational success rate, the success rate of the scaffolds generated through Rosetta-based protocols is usually highly correlated with the protein fold type. The helical proteins usually have significantly higher solubility rate than β -sheet containing proteins. The helical repeat proteins can have success solubility rate up to $74/83 = 89.2\%$ while the success rate for NTF2-like proteins was $17/64 = 26.6\%$ (Basanta et al, PNAS 2020), while for the new ML protocol, it was 84.4% (81/96), regardless of topology. Their solubility and folding rate are significantly higher than previous work as described in our manuscript.

The most important difference between the old and new protocol is, of course, that the new protocol can freely sample structure space and generate highly diverse folds with different secondary structure content, etc, whereas in the original protocol the secondary structure composition had to be precisely specified.

Page 4, line 42:

Compared to previous Rosetta-based scaffold generation pipelines, our pipeline can freely sample structure space and generate widely diverse protein architectures. Our pipeline also has significantly higher efficiency at both the computational and experimental levels. 21,021 out of roughly 28,000 (73.84%) generated pseudocycles had Rosetta, AF2, and RF metrics predictive of folding. By comparison, for the simpler problem of helical repeat protein design, the success rate for a previous Rosetta-based protocol was only $11,243/2,880,000 = 0.39\%$. At the experimental level, 84.4% (81/96) of the designs from our pipeline are highly soluble, compared to $17/64 = 26.6\%$ for previous Rosetta based design of small molecule-binding (NTF2) scaffolds 16 (the solubility rate for Rosetta-based design of helical repeat proteins lacking cavities is $74/\beta$ repeats that close to $83 = 89.2\%$).

Page 12, line 17:

The model of all unique scaffolds (9838) and the sequences and the design models of the 96 characterized designs were uploaded to github (https://github.com/LAnAlchemist/Psedocycles_NSMB.git).

The scaffold generation scripts were uploaded to github (https://github.com/dmitropher/af2_multistate_hallucination.git).

Further, how do AlphaFold2 and RoseTTAfold compare. Why were there 5 models with AF2 done but only one with RoseTTAfold?

Because DeepMind has a very large amount of compute resources, they were able to train 5 completely independent models, whereas we were able to train only a single version of RoseTTAfold. Each training procedure requires many weeks on large numbers of GPUs.

The 5 models generated by AF2 and the single model generated by RF generally agree with each other in terms of the quality of the designed scaffold (Figure S4).

Experimentally, it is good to see that many proteins can be expressed – however the authors should be careful in their description. Reading the methods section, it appears that in the first step they test protein expression, but not whether the proteins are found in the soluble fraction of the cell extract. If this is true it should be described as ‘can be expressed’ and not as ‘soluble’. This would also explain why less proteins can be purified.

We apologize for lack of clarity and will be more precise in the revised manuscript. In the first step protein expression test, we screened for protein from the soluble fraction using SDS-PAGE gel. We have edited the text to make sure this information is clearly delivered.

Page 14, line 19, original:

The lysed cells were spun down and 10 uL of clear supernatant was checked for each protein on SDS-PAGE gel.

Page 14, line 19, after edit:

The lysed cells were spun down and 10 uL of clear supernatant was checked for each protein on SDS-PAGE gel for indication of protein solubility from the soluble fraction.

I appreciate that the SEC traces are shown, a comparison of expected versus apparent molecular weight would be desirable though as to judge oligomerization states. The

authors speak of monodispersity in samples, was light scattering used to test this?

Light scattering was used for representative samples analyzed with SEC-MALS (Size Exclusion Chromatography with Multi-Angle Light Scattering). The thirteen examples provided in the main text all centered around the expected monomeric retention volume (Figure 2). We also provided the SAXS data for the thirteen examples from Figure 2, and their SAXS signals all correspond to monomeric distribution with expected fold (Figure S11). For the rest of the samples, expected retention volumes for multimeric forms of our molecules were notably different from monomeric expected retention volume. The expected monomer peak was marked out with stars on all SEC traces (Figure S10).

The solution of two crystal structures is certainly nice, even if a four-helix bundle with 2-fold symmetry is not the most exciting example one might expect. With the large cavity between the two α -hairpin it would be nice to show also the contacts that contribute to the seemingly high stability of H12.

We added one more crystal structure which has a larger pocket than H12 and H10.

I write seemingly as there are details missing on the equilibration before taking the CD spectra at different temperatures. To be able to view a T-melt trace would be nice and might be more informative. This is especially interesting as the authors argue that native proteins are not stable enough to introduce new binding or active sites. But how stable are the new proteins in comparison?

The CD spectra were scanned at 25 °C, 55 °C, 95 °C, and final scan was performed at 25 °C after slow temperature drop (Figure 2 and Figure S7). The temperature ramping was very slow and most of our proteins showed very little unfolding. We included these unfolding traces in the supplement (Figure S8-9), however, the magnitude of unfolding for most of the designs were quite small.

The docking results are also a bit confusing. In figure 4a for LFN there is a docking with the pseudocycle structure, but in b and c only docking results with natives and NTF2 are listed.

Figure 4a shows the best-scored binders designed using only pseudocycles. Figure 4b-c shows the best-scored binders designed using pseudocycles, NTF2, and native proteins.

Overall, this manuscript is not ready for publication and the claims are unfortunately not supported sufficiently.

This completely disagrees with the assessment of the first two reviewers. What claims made in the paper are not supported sufficiently?

Additionally, we added one more crystal structure of the designed pseudocycles, which further supported our conclusions that our pipeline can generate de novo cyclic proteins with pockets

inside with high accuracy. We added this structural data and related analysis to the manuscript.

Decision Letter, first revision:

Message: Our ref: NSMB-A46461B

9th Mar 2023

Dear Dr. An,

Thank you for submitting your revised manuscript "Hallucination of closed repeat proteins containing central pockets" (NSMB-A46461B). It has now been seen by the original referees and their comments are below. You will see that while Reviewer #3 reiterates several of their outstanding concerns regarding experimental validation of ligand binding to the designed pockets, the other reviewers find that the paper has improved in revision. As discussed in the previous round of revision, editorially, we feel that the validation presented in the study is sufficient and that additional experimental validation of the ligands bound to the designed pockets may be outside the scope of the current work, and therefore, we'll be happy in principle to publish it in Nature Structural & Molecular Biology, pending minor revisions to satisfy the referees' other final minor requests and to comply with our editorial and formatting guidelines.

We are now performing detailed checks on your paper and will send you a checklist detailing our editorial and formatting requirements in a couple of weeks. Please do not upload the final materials and make any revisions until you receive this additional information from us.

To facilitate our work at this stage, it is important that we have a copy of the main text as a word file. If you could please send along a word version of this file as soon as possible, we would greatly appreciate it; please make sure to copy the NSMB account (cc'ed above).

Sincerely,
Sara

Sara Osman, Ph.D.
Associate Editor
Nature Structural & Molecular Biology

Reviewer #1 (Remarks to the Author):

The authors have provided sufficient data or added rational explanations for all the points I concerned.

Reviewer #2 (Remarks to the Author):

The revision has addressed the points raised during the review of the original manuscript and I recommend it now for publication.

Reviewer #3 (Remarks to the Author):

The authors addressed a number of points by all reviewers, in particular the plan to publish details on the code and the pseudocycle structures on github upon publication is an important improvement. It seems however that my main concern was not clear enough as it is unfortunately not addressed:

While the manuscript describes a nice new pipeline for the design of pseudocycles it does not become clear that the new scaffolds are really better suited for ligand binding and enzyme design as is the claim clearly stated at the end of the abstract as well as throughout the manuscript. If this was shown I would see the manuscript fit for a journal such as NSMB that places a focus on functional and mechanistic understanding. Docking results might hint at more possibilities for placing ligands when using certain scoring functions but a real experimental test for even a small set is missing and currently rejected by the authors.

Already the paper by Basanta et al 2020 in PNAS (ref 16) was claiming more diverse pocket structures ready for ligand binding and enzyme design. With the new pipeline that nicely combines already published computational methods, an impressive set of pseudocycle structures has been generated – data that is surely worthwhile publishing in a more specialized journal. The next step that needs to be taken and that the authors use as their motivation is ligand binding design, but this has not been shown here any differently than it has been done by Basanta et al or in any follow up of the PNAS paper. The measurement of ligand binding for a few chosen examples is not much work but such data would really be a step forward and a true test. Without such proof the claim that these pseudocycles are better starting points for design of function should not be made.

In addition, the comparison of designs at the experimental level (in terms of soluble expression) needs clarification. In Baratas et al the designs are screened differently, first a HTP screen for stability by YSD and then more detailed on a focused set, which led to 7 soluble well folded proteins out of 17 (41%). Now this is still much lower than the 81 out of 96 presented here, but it should not be neglected that the new pipeline uses ProteinMPNN, which has been already published by Dauparas et al 2022 in Science (ref 10) to improve success rates dramatically namely to 73 out of 96 tested. The statement “Their solubility and folding rate are significantly higher than previous work as described in our manuscript” can thus not be made.

As an aside, the following point was not addressed:

The solution of two crystal structures is certainly nice, [...]. With the large cavity between the two α -hairpin it would be nice to show also the contacts that contribute to the seemingly high stability of H12.

The question here was not simply for another structure, while this is surely a valuable addition, but an encouragement to learn more from the structure of the designs. Why not

explore and showcase the structural features better. In H12 there are probably only a few contacts that support the high stability of the helical bundle as the cavity appears fairly large for such a small scaffold, at least in the shown picture.

Author Rebuttal, first revision:**Response to reviewers**

We thank the reviewers for their feedback. In the responses below, the original reviewers' comments are italicized and in bold. The responses are not italicized and follow each question. Edits of the manuscript are italicized, and the page number and the line number are provided where relevant. The revised sentences are colored yellow in the revised version of the manuscript.

Reviewer #1:**Remarks to the Author:**

The authors have provided sufficient data or added rational explanations for all the points I concerned.

We thank the reviewer for providing all the suggestions which helped us to improve the manuscript.

Reviewer #2:**Remarks to the Author:**

The revision has addressed the points raised during the review of the original manuscript and I recommend it now for publication.

We thank the reviewer for providing all the suggestions which helped us to improve the manuscript.

Reviewer #3:**Remarks to the Author:**

The authors addressed a number of points by all reviewers, in particular the plan to publish details on the code and the pseudocycle structures on github upon publication is an important improvement. It seems however that my main concern was not clear enough as it is unfortunately not addressed:

While the manuscript describes a nice new pipeline for the design of pseudocycles it does not become clear that the new scaffolds are really better suited for ligand binding and enzyme design as is the claim clearly stated at the end of the abstract as well as throughout the manuscript. If this was shown I would see the manuscript fit for a journal such as NSMB that places a focus on functional and mechanistic understanding. Docking results might hint at more possibilities for placing ligands when using certain scoring functions but a real experimental test for even a small set is missing and currently rejected by the authors.

The NSMB is a great “integrated forum for structural and molecular studies”, according to the journal aims. In this manuscript, we described how to sample proteins with pockets in detail and provided related molecular biology, structural biology data on showing the success of this newly developed pipeline, and we did propose the potential application of the results of this pipeline will be small molecule binder design, which currently is in progress. We feel this content fits well with the aim of the NSMB journal.

Already the paper by Basanta et al 2020 in PNAS (ref 16) was claiming more diverse pocket structures ready for ligand binding and enzyme design.

To the diversity difference between the pseudocycle paper and the NTF2 paper, it is clear that the pseudocycles contains 9838 sets of scaffolds which each of them is a different fold type and contains a unique pocket shape. On the contrary, the NTF2 paper described a great way to generate the NTF2s, which is one single type protein fold and pocket shape. NTF2s definitely provided a great pocket diversity compared to the studies before 2020, the pseudocycles, however, provided a significantly bigger scaffold diversity than the NTF2s.

With the new pipeline that nicely combines already published computational methods, an impressive set of pseudocycle structures has been generated – data that is surely worthwhile publishing in a more specialized journal. The next step that needs to be taken and that the authors use as their motivation is ligand binding design, but this has not been shown here any differently than it has been done by Basanta et al or in any follow up of the PNAS paper.

As we explained in the previous response and the above two paragraphs, the binder design and the protein design problem focus on very different things, which are not suitable to put in the same manuscript.

In the pocket-containing protein design project (a.k.a this manuscript), we focused on sampling the pocket space, and performed high-quality scaffold and sequence design for these proteins which have giant holes built in them. This problem solving procedure required a new protein backbone sampling pipeline together with new protein design methods.

The ligand binding problem is in fact, a harder and a totally different problem comparing to the protein scaffold design problem, it needs to solve three completely different and challenging problems: 1) ligand-protein scaffold shape complementary sampling 2) ligand docking conformation sampling; 3) high-accuracy ligand-protein interface sequence design. All these problems require new design tools and pipelines which are significantly different from the protein structural design pipeline, and will significantly distract the contents which we are trying to describe here. Considering the manuscript will be extremely different if including ligand binding as explained above, we do not think it is suitable to discuss ligand binding problems here.

The measurement of ligand binding for a few chosen examples is not much work but such data would really be a step forward and a true test. Without such proof the claim that these pseudocycles are better starting points for design of function should not be made.

The computational and laboratorial experiments for the ligand binding problem will be significantly more than the protein scaffold design problem. That is to say, it is anything but ***not much work***, and actually really requires an independent story for detailed explanation.

In addition, the comparison of designs at the experimental level (in terms of soluble expression) needs clarification. In Baratas et al the designs are screened differently, first a HTP screen for stability by YSD and then more detailed on a focused set, which led to 7 soluble well folded proteins out of 17 (41%). Now this is still much lower than the 81 out of 96 presented here, but it should not be neglected that the new pipeline uses ProteinMPNN, which has been already published by Dauparas et al 2022 in Science (ref 10) to improve success rates dramatically namely to 73 out of 96 tested. The statement "Their solubility and folding rate are significantly higher than previous work as described in our manuscript" can thus not be made.

The scaffold generation method and sequence design methods are both different from the NTF2 paper, but both of them perform significantly better than the methods used in the NTF2 paper

(see discussion part). Considering we do include the proteinMPNN redesign as a step in the pipeline, we do not see why we cannot claim "***Their solubility and folding rate are significantly higher than previous work as described in our manuscript***".

As an aside, the following point was not addressed:

The solution of two crystal structures is certainly nice, [...]. With the large cavity between the two a-hairpin it would be nice to show also the contacts that contribute to the seemingly high stability of H12.

The question here was not simply for another structure, while this is surely a valuable addition, but an encouragement to learn more from the structure of the designs. Why not explore and showcase the structural features better. In H12 there are probably only a few contacts that support the high stability of the helical bundle as the cavity appears fairly large for such a small scaffold, at least in the shown picture.

We apologize for forgetting to include this information in our first response, please see Figure S10 for this information.

Original:

Page 3, second to last line:

The outside short helix stubs form hydrophobic interfaces with the neighboring two long helices, which stabilize the helical pocket.

New version:

Page 3, second to last line:

*The outside short helix stubs form hydrophobic interfaces with the neighboring two long helices, and the hydrophobic side chains lock the helices while still leaving a big pocket in the middle (see **Supplementary Figure. S10**).*

Final Decision Letter:

Message 28th Aug 2023

:

Dear Dr. An,

We are now happy to accept your revised paper "Hallucination of closed repeat proteins containing central pockets" for publication as a Article in Nature Structural & Molecular

Biology.

As soon as your article is published, you can generate your shareable link by entering the DOI of your article here: <http://authors.springernature.com/share> `http://authors.springernature.com/share`. Corresponding authors will also receive an automated email with the shareable link

Your paper will be published online soon after we receive proof corrections and will appear in print in the next available issue. You can find out your date of online publication by contacting the production team shortly after sending your proof corrections. Content is published online weekly on Mondays and Thursdays, and the embargo is set at 16:00 London time (GMT)/11:00 am US Eastern time (EST) on the day of publication. Now is the time to inform your Public Relations or Press Office about your paper, as they might be interested in promoting its publication. This will allow them time to prepare an accurate and satisfactory press release. Include your manuscript tracking number (NSMB-A46461C) and our journal name, which they will need when they contact our press office.

About one week before your paper is published online, we shall be distributing a press release to news organizations worldwide, which may very well include details of your work.

We are happy for your institution or funding agency to prepare its own press release, but it must mention the embargo date and Nature Structural & Molecular Biology. If you or your Press Office have any enquiries in the meantime, please contact press@nature.com.

Please note that *Nature Structural & Molecular Biology* is a Transformative Journal (TJ). Authors may publish their research with us through the traditional subscription access route or make their paper immediately open access through payment of an article-processing charge (APC). Authors will not be required to make a final decision about access to their article until it has been accepted. [Find out more about Transformative Journals](https://www.springernature.com/gp/open-research/transformative-journals)

Authors may need to take specific actions to achieve [compliance with funder and institutional open access mandates](https://www.springernature.com/gp/open-research/funding/policy-compliance-faqs). If your research is supported by a funder that requires immediate open access (e.g. according to [Plan S principles](https://www.springernature.com/gp/open-research/plan-s-compliance)) then you should select the gold OA route, and we will direct you to the compliant route where possible. For authors selecting the subscription publication route, the journal's standard licensing terms will need to be accepted, including [self-archiving policies](https://www.springernature.com/gp/open-research/policies/journal-policies). Those licensing terms will supersede any other terms that the author or any third party may assert apply to any version of the manuscript.

Sincerely,
Sara

Sara Osman, Ph.D.
Associate Editor
Nature Structural & Molecular Biology

Click here if you would like to recommend Nature Structural & Molecular Biology to your librarian:

<http://www.nature.com/subscriptions/recommend.html#forms>